# Opposing p53 and mTOR/AKT promote an in vivo switch from apoptosis to senescence upon telomere shortening in zebrafish

**Mounir El Maï**[1,2†]**, Marta Marzullo**[2†]**, Inês Pimenta de Castro**[2†]**, Miguel Godinho Ferreira**[1,2*]

[1]Institute for Research on Cancer and Aging of Nice (IRCAN), Université Côte d'Azur, Nice, France; [2]Instituto Gulbenkian de Ciência, Oeiras, Portugal

**Abstract** Progressive telomere shortening during lifespan is associated with restriction of cell proliferation, genome instability and aging. Apoptosis and senescence are the two major outcomes upon irreversible cellular damage. Here, we show a transition of these two cell fates during aging of telomerase deficient zebrafish. In young telomerase mutants, proliferative tissues exhibit DNA damage and p53-dependent apoptosis, but no senescence. However, these tissues in older animals display loss of cellularity and senescence becomes predominant. Tissue alterations are accompanied by a pro-proliferative stimulus mediated by AKT signaling. Upon AKT activation, FoxO transcription factors are phosphorylated and translocated out of the nucleus. This results in reduced SOD2 expression causing an increase of ROS and mitochondrial dysfunction. These alterations induce p15/16 growth arrest and senescence. We propose that, upon telomere shortening, early apoptosis leads to cell depletion and insufficient compensatory proliferation. Following tissue damage, the mTOR/AKT is activated causing mitochondrial dysfunction and p15/16-dependent senescence.

***For correspondence:**
Miguel-Godinho.FERREIRA@
unice.fr

[†]These authors contributed
equally to this work

**Competing interests:** The
authors declare that no
competing interests exist.

**Reviewing editor:** Dario
Riccardo Valenzano, Max Planck
Institute for Biology of Ageing,
Germany

## Introduction

Accumulation of DNA damage impairs cellular function giving rise to defects in tissue function, diseases and aging (*Jackson and Bartek, 2009*). Cells evolved DNA repair mechanisms to counteract the accumulation of DNA damage. However, when the damage persists, cells undergo cell-cycle arrest, resulting in either apoptosis or senescence (*Childs et al., 2014*).

Apoptosis constitutes a highly regulated pathway that results in cell death, in a p53-dependent manner. In order to maintain organ homeostasis, apoptotic cells are eliminated from the tissue and replaced by dividing cells in a compensatory mechanism (*Fogarty and Bergmann, 2017*). Senescence is a permanent cell-cycle arrest, generally associated with pro-inflammatory events, and with a secretome known as Senescence Associated Secretory Phenotype (SASP) (*Coppé et al., 2008*). In contrast to apoptosis, senescent cells are eliminated by the immune system. Inefficient removal of senescent cells results in their accumulation and persistent SASP production. This phenomenon was previously associated with aging and age-related phenotypes (*Krishnamurthy et al., 2004*).

The CDK inhibitor (CKI) p16 encoded by the INK4a/ARF locus (CDKN2A), is a tumor suppressor gene that limits cell proliferation and is associated with cell senescence (*Liao and Hung, 2003*). It belongs to the INK family of CKIs that includes p15INK4b, p18INK4c and p19INK4d (*Kamb, 1995*; *Vidal and Koff, 2000*). The comparisons of the INK4a/ARF gene structure between human, mouse, chicken and the fugu fish revealed a dynamic evolutionary history of this locus (*Gilley and Fried, 2001*; *Kim et al., 2003*). p15INK4b (p15), a close relative of p16, is encoded by the INK4b

gene (CDKN2B), located immediately upstream of the INK4a/ARF locus. Similar to p16, p15 is involved in cell senescence (*Fuxe et al., 2000*; *Hitomi et al., 2007*; *Senturk et al., 2010*). In fugu fish, as in chicken, p16INK4a is mutated and does not encode a functional protein. Thus, fugu and chicken INK4a loci express Arf but not p16 (*Liao and Hung, 2003*). Consequently, the authors proposed that, in these animals, the function of p16 was evolutionarily replaced by p15.

Under regular physiological conditions, cells with a high turn-over rate (including epithelial and germinal cells) preserve tissue homeostasis by undergoing continuous apoptosis and compensatory cell proliferation. However, in some animals including humans, cells can only go through a finite number of divisions before entering irreversible cell-cycle arrest, a process known as replicative senescence. Telomere erosion was proposed to constitute the 'molecular clock' that determines the number of divisions a cell can undergo before reaching senescence, a phenomenon known as Hayflick limit (*Bodnar et al., 1998*; *Hayflick, 1965*). In most eukaryotes, telomere shortening is counteracted by telomerase, though its expression is restricted in most human somatic cells (*Forsyth et al., 2002*). Consequently, telomeres shorten significantly during human aging (*Aubert and Lansdorp, 2008*).

Vertebrate telomerase mutant animal models have been used to investigate the direct association between telomere shortening and tissue dysfunction. Even though the common lab mouse has 5-10x longer telomeres than humans, upon several generations of in-crossing (G4-6), telomerase-deficient mice demonstrated premature features of aging, including reduced cell proliferation and increased apoptosis in several tissues (*Lee et al., 1998*; *Rudolph et al., 1999*). As observed in humans, telomere length in zebrafish ranges between 5–15 Kb, and telomerase is restricted in somatic tissues (*Anchelin et al., 2011*; *Anchelin et al., 2013*; *Carneiro et al., 2016b*; *Henriques et al., 2013*). As a consequence, we showed that telomeres shorten throughout zebrafish natural aging (*Carneiro et al., 2016b*). First-generation adult zebrafish telomerase mutants (*tert-/-*) exhibit short telomeres and die as young adults. They display premature aging phenotypes including reduced cell proliferation, loss of tissue homeostasis and functional organ decline (*Anchelin et al., 2013*; *Carneiro et al., 2016b*; *Henriques et al., 2013*). Strikingly, we observed that telomeres in both aged wild-type and *tert-/-* zebrafish reach a similar length as they exhibit aging phenotypes (*Carneiro et al., 2016b*). Accumulation of DNA damage, decline in cell division and organ dysfunction are associated with tissue-dependent telomere shortening (*Anchelin et al., 2013*; *Carneiro et al., 2016b*; *Henriques et al., 2013*). Likewise, old age afflictions including infertility, infections, cachexia and cancer are accelerated in young telomerase mutant zebrafish (*Carneiro et al., 2016b*). Similar to humans affected by telomeropathies (*Opresko and Shay, 2017*), young zebrafish telomerase mutants display phenotypes of old age, including 'genetic anticipation', in which second generation telomerase deficient animals have aggravated phenotypes and die as larva (*Henriques et al., 2013*; *Anchelin et al., 2013*). Overall, telomeres of naturally aged zebrafish shorten to critical lengths and this phenomenon is related with age-associated dysfunction and diseases. Because, like humans, telomere shortening is part of physiologic aging, zebrafish constitutes an appropriate vertebrate model to study the consequences of short telomeres in aging (*Carneiro et al., 2016a*).

As telomeres become critically short, they accumulate γH2A.X and activate the DNA Damage Responses (DDRs) (*d'Adda di Fagagna et al., 2003*). One of the mediators of DDR is the onco-suppressor p53, which accumulates upon telomere shortening and may result in either cell senescence or apoptosis (*Li et al., 2016*). The signals leading to each cell fate in response to p53 accumulation are unclear to date. Previous studies suggested that cellular senescence is associated with increased levels of mTOR/AKT signaling (*Miyauchi et al., 2004*; *Moral et al., 2009*; *Leontieva and Blagosklonny, 2013*). AKT is a serine/threonine protein kinase that is activated upon pro-proliferative extracellular signals. mTOR/AKT pathway is triggered by growth factor receptors, including the Insulin Growth Factor Receptor (IGFR) (*Liao and Hung, 2010*). Activation of AKT- and mTORC2-mediated phosphorylation results in the phosphorylation of the forkhead transcription factors, FoxO1 and FoxO4 (*Tuteja and Kaestner, 2007*). Once phosphorylated, the FoxO family proteins translocate outside the nucleus, followed by repression of their main target genes, including mitochondrial superoxide dismutase SOD2. Prolonged and uncontrolled activation of this pathway results in mitochondrial dysfunction and increased ROS levels (*Nogueira et al., 2008*).

In our study, we investigated the in vivo switch between apoptosis and cell senescence as a consequence of telomere shortening in *tert-/-* zebrafish. We describe that early in life, telomerase

deficiency results in p53 mediated apoptosis and loss of tissue homeostasis, causing pro-proliferative AKT pathway activation in older individuals. AKT/FoxO signal cascade then triggers a switch that results in mitochondrial dysfunction, increased levels of ROS and p15/16, leading to cell senescence.

## Results

### tert-/- zebrafish proliferative tissues undergo a time-dependent switch from apoptosis to senescence

Apoptosis is a process in which programmed cell death allows for clearance of damaged cells (*Hawkins and Devitt, 2013*). In contrast, replicative senescence is a state of terminal proliferation arrest associated with irreparable DNA damage, such as critically short telomeres resulting from cell division (*Olovnikov, 1973*; *Shay and Wright, 2000*). To explore the molecular mechanisms underlying the cell-fate decision between apoptosis and senescence, we used telomere attrition as a trigger of these two possible outcomes.

First-generation *tert-/-* zebrafish have shorter telomeres than their wild-type (WT) siblings, develop several degenerative conditions affecting mainly highly proliferative tissues, such as the testis and gut, and die prematurely (*Anchelin et al., 2013*; *Carneiro et al., 2016b*; *Henriques et al., 2013*). At 3 months of age, *tert-/-* fish are macroscopically similar to their WT siblings, with gut and testis being histologically indistinguishable from WT (*Figure 1A*). However, at this early age, we previously observed that average telomere length in *tert-/-* mutants is shorter than WT and triggers the onset of DDR and increased apoptosis (*Carneiro et al., 2016b*).

We analyzed the presence of apoptotic cells in 3-month-old *tert-/-* gut and testis using the TUNEL assay. We confirmed that, even in the absence of macroscopic defects, *tert-/-* gut and testis exhibit a higher number of apoptotic TUNEL-positive cells when compared to their WT siblings (*Figure 1C, E and F*; WT N = 3–6, *tert-/-* N = 3–6, p<0.05). In order to confirm the activation of DDR, we analyzed the phosphorylation levels of the DNA damage marker γH2A.X in whole cell lysates from gut and testis of 3-month-old *tert-/-* zebrafish (*Figure 2A*). We detected a significant increase of the ATM-dependent phosphorylated form of γH2A.X in ser139 in both 3-month-old *tert-/-* gut and testis (*Figure 2A*; WT N = 8–14, *tert-/-* N = 5–15, p<0.05). As expected, we observed a concomitant increased level of p53 in both tissues of *tert-/-* zebrafish (*Figure 2A*; WT N = 8–13, *tert-/-* N = 8–13, p<0.05).

In light of the differences found in the INK4a/ARF locus of zebrafish and mammals, we tested the conservation of this protein and the validity of the mammalian anti-p16 antibody (sc-1661, Santa Cruz Biotechnology) used for senescence analysis. To this purpose, we designed antisense morpholino oligonucleotides (MOs) against cdkn2a/b (hereafter p15/16)) and injected increasing amounts in 1 cell-stage embryos (*Figure 1—figure supplement 1*). An unrelated control morpholino sequence was included as negative control (CTR MO). Upon morpholino injection, larvae were collected at 3 days post-fertilization (dpf) and tested for the expression of the p15/16 protein. Western Blot analysis revealed that injection of an increasing concentration of p15/16 MOs causes a reduction in the amount of protein recognized by the anti-p16 antibody, thus indicating that the sequence of the protein associated with senescence is conserved from mammals to zebrafish (*Figure 1—figure supplement 1*).

Strikingly, even though DDR is active in 3-month-old *tert-/-*, the tissues analyzed did not exhibit signs of cell senescence. We were unable to detect senescence-associated beta-galactosidase (SA-β-Gal) activity in both gut and testis of *tert-/-* fish (*Figure 1C*). Consistently, we observed no differences in the expression of p15/16 in 3-month-old WT and *tert-/-* zebrafish neither by immunofluorescence staining (*Figure 1C, E and F*; WT N = 3–5, *tert-/-* N = 3–5), RT-qPCR nor western blot (*Figure 2A and C*; WT N=-6, *tert-/-* N = 5–6), with the exception of lower p15/16 levels by western blot of *tert-/-* testis. Therefore, in 3-month-old *tert-/-* fish, a stage in which tissue integrity is preserved, telomere-dependent DDR signaling induces apoptosis but no detectable cell senescence.

To investigate the consequences of telomere erosion and chronic DDR activation through aging, we analyzed the gut and testis of older *tert-/-* animals (6–9 month of age). Contrary to what was observed in 3-month-old fish, older *tert-/-* zebrafish exhibit morphological tissue defects (*Figure 1B*), resulting in width lengthening of the gut *lamina propria* and testis atrophy (as previously observed by us, *Henriques et al., 2013*).

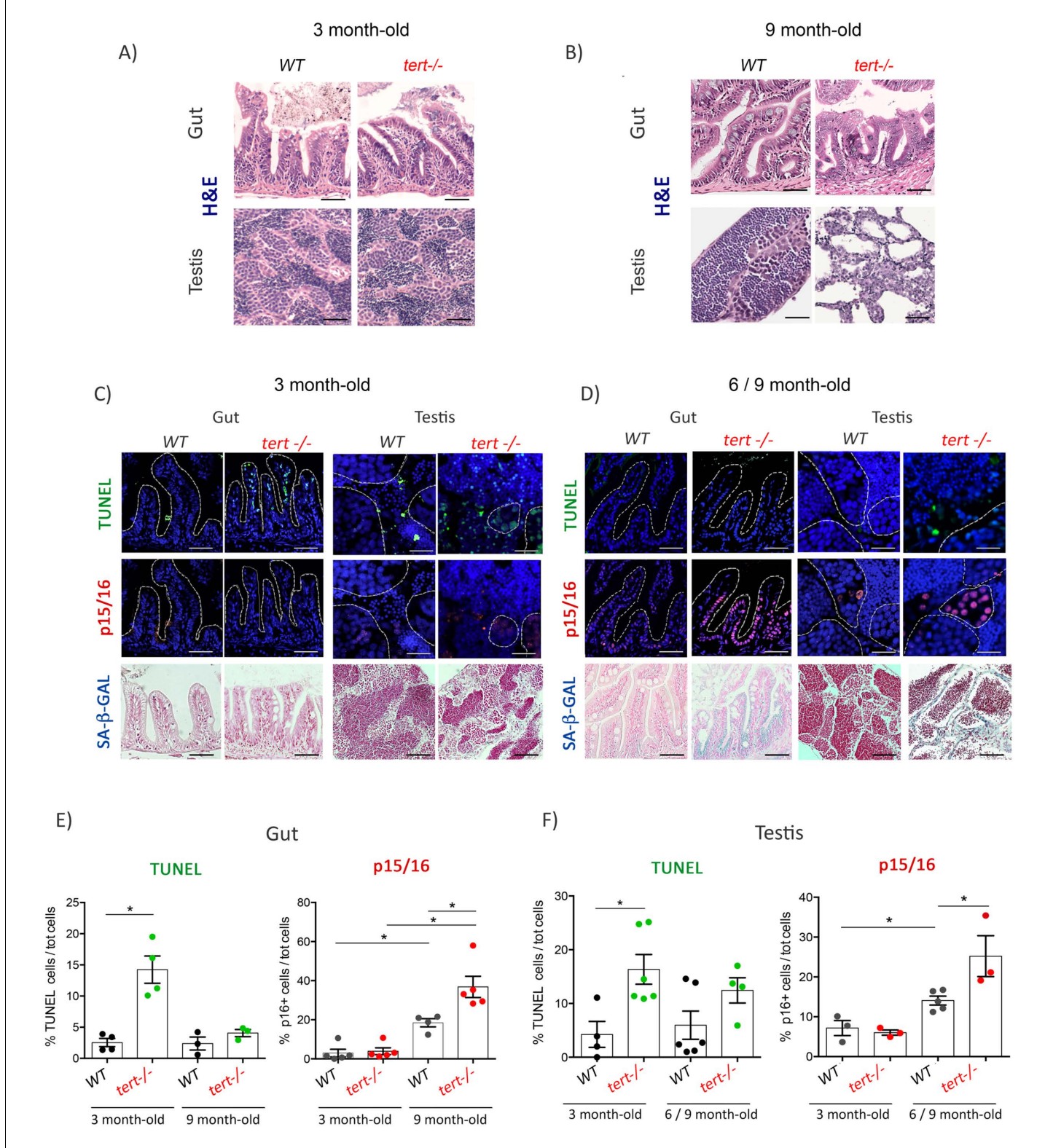

**Figure 1.** Proliferative tissues of *tert-/-* zebrafish undergo an in vivo switch from apoptosis to senescence with age. (A-B) Representative haematoxylin and eosin-stained sections of gut (scale bar: 40 µm) and testis (scale bar: 25 µm) from 3-month-old (**A**) or 9-month-old (**B**) of WT and *tert-/-* siblings. While no macroscopic tissue defects are distinguishable at 3 months (N = 3), 9 month *tert-/-* (N = 3) exhibit altered gut and testis structures. (C-D) Representative immunofluorescence images of apoptosis (TUNEL) or senescence (p15/16 and SA-β-GAL) of gut and testis from 3 month (**C**) or 6–9 month-old (**D**) WT and *tert-/-* siblings (N = 3–6 each)(scale bar: 25 µm). Dashed outlines locate cysts of spermogonia cells or spermatocytes (testis) or

*Figure 1 continued on next page*

*Figure 1 continued*

villi (gut). At 3 months, both tissues show an increased number of apoptotic cells in *tert-/-* compared to WT. At that age, no signs of senescence are visible in these tissues. However, senescent cells appear in the gut and testis of 6–9 month-old *tert-/-* fish depicting a switch between apoptosis and senescence. (**E-F**) Quantification of the percentage of TUNEL and p15/16 positive cells in 3 month and 6–9 month-old *tert-/-* or WT. Data are represented as mean ± SEM. * p-value<0.05; using the Mann-Whitney test.

The online version of this article includes the following source data and figure supplement(s) for figure 1:

**Source data 1.** Quantification of TUNEL and p16 positive cells in gut, as plotted in *Figure 1E*.
**Source data 2.** Quantification of TUNEL and p16 positive cells in testis, as plotted in *Figure 1F*.
**Figure supplement 1.** Anti-p16 antibody validation in zebrafish through antisense morpholino knock-down of *cdkn2a/b* (p15/16).
**Figure supplement 2.** Bcl-XL is overexpressed in 9 month but not 3-month-old *tert-/-*.
**Figure supplement 2—source data 1.** Real-time qPCR data of Bcl-XL, as plotted in *Figure 1—figure supplement 2*.
**Figure supplement 3.** Apoptosis and senescence cell fate is present in the same cell types of gut and testis.

Because telomere shortening is known to induce both apoptosis and cell senescence, we hypothesized that the decline in tissue homeostasis represented a change in cell fate. Considering the proportion of 9-month-old fish exhibiting high levels of testis atrophy, we combined 6- and 9-month-old testis samples in which the extent of tissue degeneration would not affect our analysis. While *tert-/-* zebrafish showed higher apoptotic levels at 3 months than WT controls, we were unable to detect differences between *tert-/-* and WT at older ages (*Figure 1C, D, E and F*; WT N = 3–5, *tert-/-* N = 3–5). In contrast, 6- to 9-month-old *tert-/-* gut and testis exhibited a decline in apoptosis when compared to 3-month-old fish (*Figure 1E and F*; WT N = 3–6, *tert-/-* N = 3–5; p<0.05). These older tissues exhibited a clear accumulation of senescent cells in *tert-/-* zebrafish when compared to WT, as revealed by SA-β-Gal staining (*Figure 1D*). Increased senescence in 9-month-old *tert-/-* fish was confirmed by an increase in *cdkn1a* (p21) mRNA levels (*Figure 2C*; WT N = 4–6, *tert-/-* N = 4–6, p<0.05) and an increase in cdk2n2a/b (p15/16) expression observed by immunofluorescence (*Figure 1D, E and F*; WT N = 3–5, *tert-/-* N = 3–5; p<0.05), mRNA and protein levels (*Figure 2B and C*; WT N = 4–9, *tert-/-* N = 4–9; p<0.05). Consistently, we observed that reduction of apoptosis and increase of senescence were concomitant with higher levels of expression of Bcl-XL mRNA, suggesting activation of anti-apoptotic pathways in old *tert-/-* fish (*Figure 1—figure supplement 2*; WT N = 5–6, *tert-/-* N = 5–6; p<0.05).

Unexpectedly, we observed that older WT animals displayed a higher number of senescent cells with p15/16 staining in both organs at 6–9 months than at 3 months of age (*Figure 1C, D, E and F*; WT N = 3–5, *tert-/-* N = 3–5; p<0.05). Consistent with an increase of p15/16, we observed reduced differences in p53 and phospho-H2AX levels between WT and *tert-/-* at later ages (*Figure 2B*; WT N = 6, *tert-/-* N = 9). Differences in p53 must be still effective, however, since we detected higher *cdkn1a* (p21) mRNA levels in both gut and testis of *tert-/-* mutants (*Figure 2C*; WT N = 4–7, *tert-/-* N = 4–6; p<0.05). Considering that the average lifespan of zebrafish is 3.5 years and a reproductive decline around 18 months of age (*Carneiro et al., 2016b*), these results suggest that it is possible to detect early signs of aging in adult WT individuals.

In order to determine which cell types were affected by apoptosis and senescence throughout aging, we scrutinized the immunofluorescence images of both gut and testis of *tert-/-* individuals at different ages. We noticed that at 3 months of age, enterocytes in the gut and A and B spermatogonia in testis represented the majority of cells exhibiting higher levels of apoptosis, as observed by TUNEL (*Figure 1—figure supplement 3A–B*). Strikingly, these same cell types displayed higher levels of p15/16 immunofluorescence staining in older *tert-/-* animals (*Figure 1—figure supplement 3A–B*). Taken together, our results provide evidence for an in vivo switch from apoptosis to senescence during aging of *tert-/-* fish. This transition is closely associated with age-dependent tissue degeneration.

## ROS accumulation and mitochondrial dysfunction become apparent upon short telomere-induced senescence

Upon DNA damage, cells initially respond by halting cell-cycle progression through a p53-mediated cell-cycle arrest (*Rodriguez and Meuth, 2006*). However, if lesions persist, expression of p16INKa increases as a consequence of mitochondrial dysfunction and ROS production (*Freund et al., 2011*; *Passos et al., 2010*). Late generation telomerase-knockout mice were suggested to cause

mitochondrial dysfunction through a p53-dependent suppression of PGC1α expression, the master regulator of mitochondrial biogenesis (*Sahin et al., 2011*). G4 mTERT deficient mice exhibit significant alterations in mitochondrial morphology, accumulation of ROS and reduced ATP generation (*Sahin et al., 2011*).

We investigated if mitochondrial dysfunction could play a part in the apoptosis-to-senescence switch observed in *tert-/-* zebrafish. First, we started by examining if p53 activation triggers the

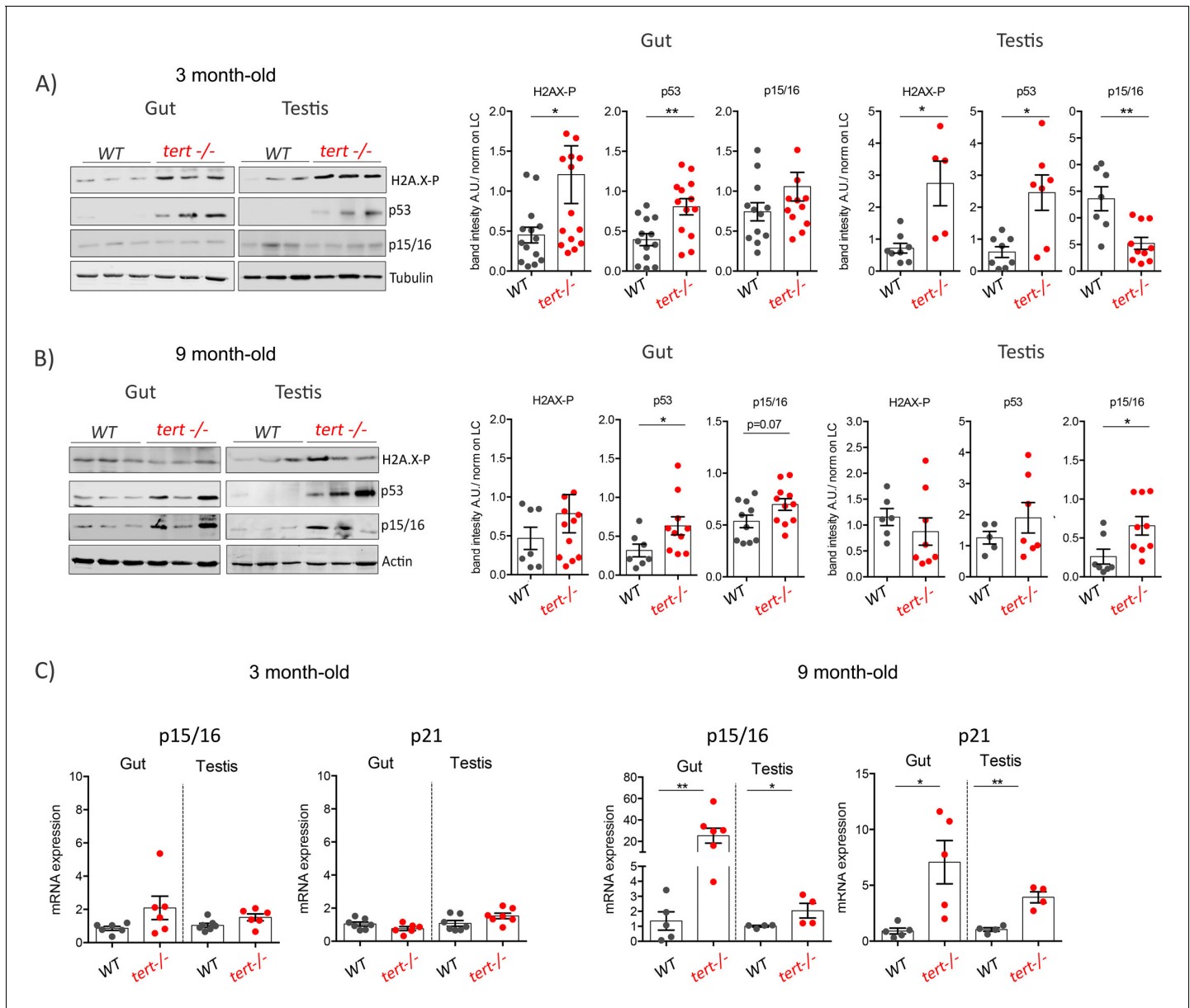

**Figure 2.** Quantification of senescence markers in gut and testis of older *tert-/-* zebrafish. (**A-B**) Western blot analysis of DNA damage and senescence-associated proteins in gut and testis of 3 month (**A**) or 9-month-old (**B**) of WT and *tert-/-* siblings (N >= 5 fish). Representative western blot (left panel) and corresponding quantification (right panel) showing induction of DNA Damage Response (H2A.X-P and p53) in 3-month-old and senescence (p15/16) in 9-month-old *tert-/-* zebrafish. (**C**) RT-qPCR analysis of senescence associated genes p15/16 and p21. RT-qPCR graphs are representing mean ± SEM mRNA fold increase after normalisation by *rpl13a* gene expression levels (* p-value<0.05; ** p-value<0.01, using the Mann-Whitney test). The online version of this article includes the following source data for figure 2:

**Source data 1.** Western Blot quantifications, as plotted in *Figure 2A-B*.
**Source data 2.** Real-time qPCR data of p15/16 and p21, as plotted in *Figure 2C*.

repression of PGC1α in zebrafish. Despite significant accumulation of p15/16 and p53 (*Figures 1F*, *2A, B and C*), we did not observe differences in neither mRNA nor protein levels of PGC1α in older *tert-/-* gut extracts (*Figure 3—figure supplement 1*; WT N = 3, *tert-/-* N = 3). However, we detected a robust increase in oxidative damage with age. By 3 months of age, the levels of ROS in *tert-/-* gut and testis did not differ significantly from their WT siblings (*Figure 3A–B*; WT N = 3, *tert-/-* N = 3). From 6 months onward, we observed a gradual and significant accumulation of ROS in both tissues in *tert-/-* relative to WT controls (*Figure 3A–B*; WT N = 3, *tert-/-* N = 3; p<0.05). Production of ROS, especially superoxide, is a necessary by-product of mitochondrial respiration (*Murphy, 2009*). Mitochondrial dysfunction is characterized by concurrent high superoxide production leading to a breakdown of membrane potential that compromises energy production and cellular metabolism (*Balaban et al., 2005*). In agreement with previous findings, we observed that gut mitochondrial morphology became increasingly rounded and swollen with the appearance of perturbed cristae in older tert-/-zebrafish (arrows, *Figure 3B*). Similarly, testis mitochondrial network became significantly fragmented (arrows, *Figure 3E*). Consistently, we observed a significant reduction of ATP levels in both tissues of *tert-/-* zebrafish (*Figure 3C–F*; WT N = 3, *tert-/-* N = 3, p<0.05). Together, our results indicate that mitochondrial function declines dramatically during aging of *tert-/-* proliferative tissues. Our results support the idea that a change in mitochondrial homeostasis may dictate the tissue's cell-fate decision.

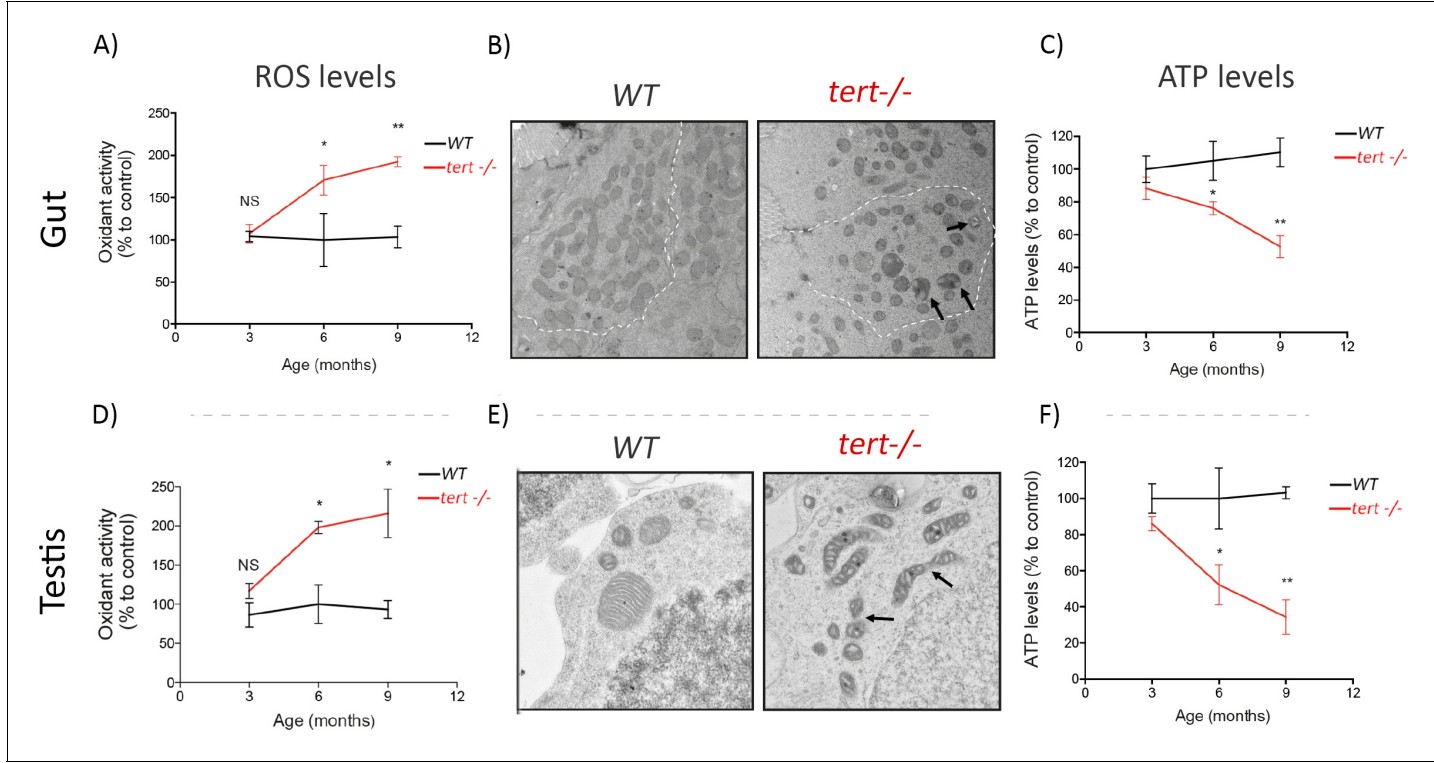

**Figure 3.** Gut and testis of *tert-/-* zebrafish are characterised by a time-dependent mitochondrial defects, increase of ROS levels and reduction of ATP levels. Gut and testis of *tert-/-* exhibit a time dependent increase in ROS levels (**A and D**) and decrease of ATP levels (**C and F**) (N >= 3 fish per time point per genotype), determined by 2′,7′-Dichlorofluorescin diacetate (DCFDA) measurement and the CellTiter-Glo Luminescent Cell Viability Assay, respectively. Representative EM images (**B and E**) of these tissues at 9 months revealed fragmented mitochondrial ultrastructure (arrows) and rounded and swollen mitochondria denoting perturbed cristae (arrows) in *tert-/-* zebrafish (N = 3 fish). Data are represented as mean ± SEM (* p-value<0.05; ** p-value<0.01, using the Mann-Whitney test).

The online version of this article includes the following source data and figure supplement(s) for figure 3:

**Source data 1.** ROS and ATP measurements, as plotted in *Figure 3*.

**Figure supplement 1.** PGC1a expression is not altered in *tert-/-* mutants compared to WT.

**Figure supplement 1—source data 1.** Real-time qPCR data of PGC1a, as plotted in *Figure 3—figure supplement 1*.

# AKT activation leads to ROS production by blocking the FoxO1/FoxO4-SOD2 molecular axis

Excessive ROS formation gives rise to oxidative stress, leading to cellular damage and, eventually, senescence (*Velarde et al., 2012*). Mitochondrial manganese superoxide dismutase (SOD2) is one of the major ROS scavengers. Notably, SOD2 expression decreases with age (*Tatone et al., 2006*; *Velarde et al., 2012*). SOD2 KO mice and connective tissue-specific SOD2 KOs have reduced lifespan and exhibit premature aging phenotypes associated with senescence but no onset of apoptosis (*Treiber et al., 2011*; *Velarde et al., 2012*).

To gain mechanistic insights into the nature of the oxidative damage observed in the *tert-/-* zebrafish, we decided to analyze the expression levels of this important antioxidant defense enzyme. Western blot analysis of 9-month-old gut and testis extracts showed a significant reduction of SOD2 protein levels in *tert-/-* mutants compared to WT (*Figure 4A* and *Figure 4—figure supplement 1*; WT N = 9–15, *tert-/-* N = 9–15, p<0.05). Consistent with our ROS results, SOD2 levels were not affected in *tert-/-* at 3 months of age (*Figure 4—figure supplement 2*; WT N = 3, *tert-/-* N = 3). These results suggest that the mechanism that copes with superoxide production inside mitochondria is compromised in older *tert-/-* mutants and, therefore, possibly responsible for the accumulation of oxidative damage in the affected tissues.

Phosphorylation (inactivation) of FoxO-family by the AKT kinase causes elevation of intracellular ROS levels through repression of detoxifying enzymes, such as SOD2 (*Brunet et al., 1999*; *Kops et al., 2002*; *Miyamoto et al., 2007*). FoxO proteins are a family of transcription factors responsible for a wide range of cellular processes, including cell cycle arrest, DNA damage response, metabolism and ROS detoxification (*Greer and Brunet, 2005*). Phosphorylation of FoxO by AKT triggers the rapid relocalization of FoxO from the nucleus to the cytoplasm, with the consequent downregulation of FoxO target genes (*Tuteja and Kaestner, 2007*). Therefore, we hypothesized that activation of AKT signaling was responsible for the increased oxidative stress in older *tert-/-* zebrafish. As expected, increased phosphorylation levels of FoxO1 and FoxO4 in the gut were correlated with lower expression levels of SOD2 (*Figure 4A*; WT N = 8, *tert-/-* N = 8, p<0.05). Phosphorylation of FoxO proteins indicates that inactivation of these transcription factors may be the underlying cause for the down-regulation of SOD2 in older *tert-/-*.

AKT is a highly conserved central regulator of growth-promoting signals in multiple cell types. The kinase activity and substrate selectivity of AKT are principally controlled by phosphorylation sites. Phosphorylation of serine 473 (pAKT-Ser473), is a consequence of activation of mammalian target of rapamycin complex 2 (mTORC2) (*Sarbassov et al., 2005*). pAKT-Ser473 is required for phosphorylation and inactivation of the FoxO transcription factors (*Guertin et al., 2006*). Accordingly, while we did not observe differences in total AKT protein levels, we detected a significant increase in the phosphorylated levels of pAKT-Ser473, denoting full activation of AKT in older but not younger *tert-/-* zebrafish (*Figure 4A* and *Figure 4—figure supplement 2*; WT N = 14, *tert-/-* N = 18, p<0.05). Therefore, AKT activation correlates with increased levels of FoxO inhibitory phosphorylation and concomitant decrease of SOD2 protein levels in *tert-/-* mutants compared to WT animals (*Figure 4A*).

In order to confirm FoxO inactivation and to correlate this event with the onset of senescence, we performed immunofluorescence co-staining of total FoxO1 with p15/16 in the gut of 9-month-old *tert-/-* and WT animals. To that intent, we measured fluorescence intensity along the enterocyte main axis and used it to calculate FoxO1 with p15/16 nuclear/cytoplasmic intensity ratios. Our results showed a striking inverse correlation between nuclear fluorescence of FoxO1 and p15/16 in senescent cells of *tert-/-* animals (*Figure 4B* red arrows, 4C, 4D and 4E, WT N = 2, *tert-/-* N = 3). According to our previous result, not all *tert-/-* enterocytes show increased p15/16 staining (*Figure 4B*, white arrows). Per cell analysis showed that *tert-/-* fish exhibit higher numbers of low nuclear FoxO1/high p15/16 cells when compared to WT (*Figure 4E*). When comparing average fluorescence intensities per fish, we observed that p15/16 nuclear/cytoplasmic levels are increased in *tert-/-* compared to WT (p<0.001), while FoxO1 nuclear staining is generally decreased in *tert-/-* fish (p=0.07, *Figure 4F*) denoting the heterogeneity of p15/16 positive cells in *tert-/-* mutants. Collectively, our results suggest that tissue damage in older animals triggers the activation of external pro-proliferative signaling. This leads to AKT-dependent inactivation of FoxO1 and FoxO4 and their translocation from the nucleus to cytoplasm, causing the down-regulation of SOD2 expression. These events

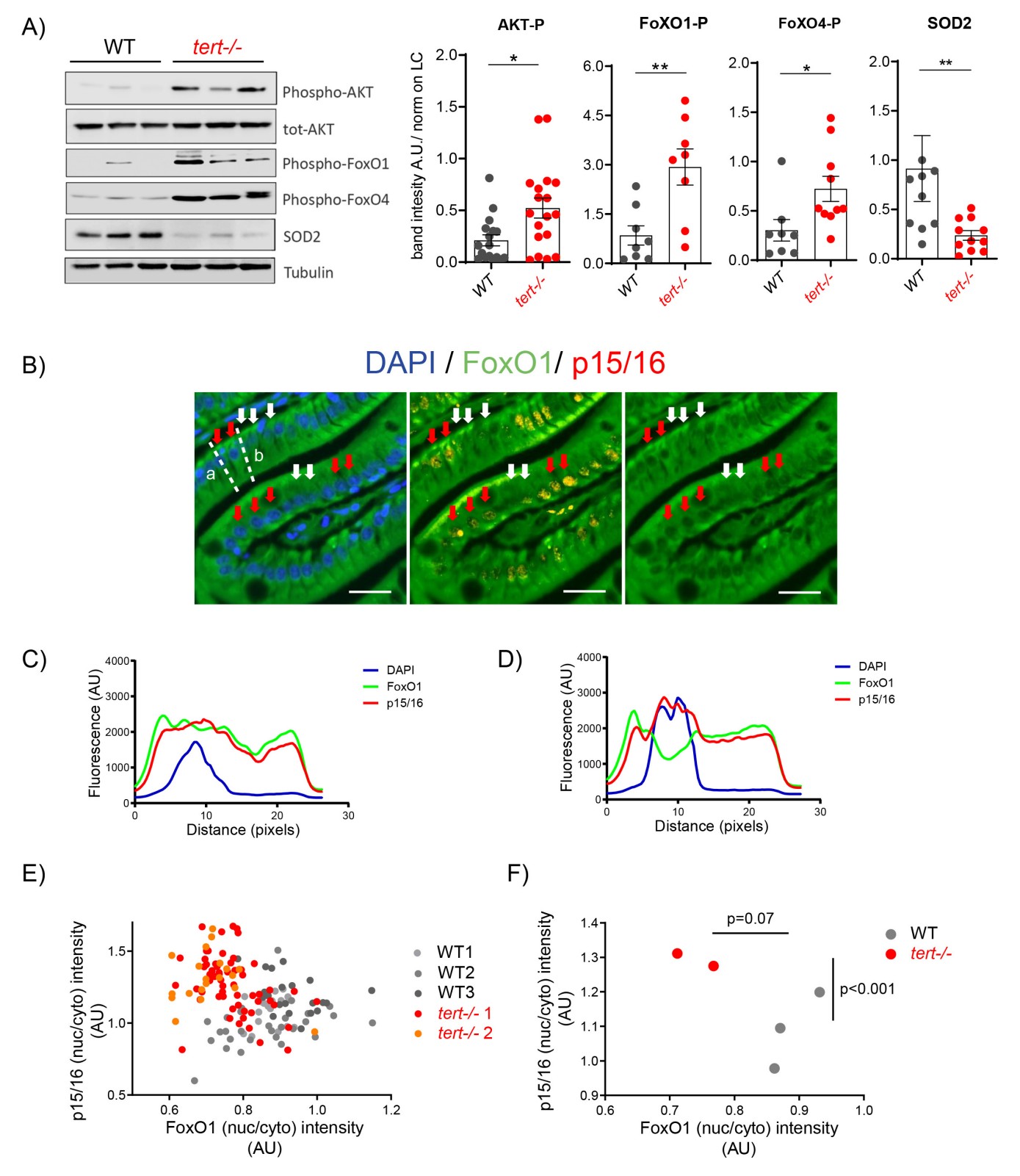

**Figure 4.** Activation of AKT in older *tert-/-* mutants results in FoxO1/4 translocation to the cytoplasm and reduction of mitochondria OxPhos defenses. (**A**) Activation of Akt leads to the inhibitory phosphorylation of FoxO1 and FoxO4 and corresponding reduction of SOD2 expression in 9-month-old *tert-/-* mutants. Western blot analysis for AKT-P, total AKT, FoxO1-P, FoxO4-P and SOD2 from gut extracts of 9-month-old *tert-/-* mutant and WT siblings (N >= 9). Representative western blot (left panel) and corresponding normalised quantification (right panel). Data are represented as

*Figure 4 continued on next page*

*Figure 4 continued*

mean ± SEM. * p-value<0.05; ** p-value<0.01 using the Mann-Whitney test. (B-F) Activation of Akt in older *tert-/-* mutant gut enterocytes leads to the translocation of FoxO1 from the nucleus to the cytoplasm and complementary accumulation p15/16 senescence marker. Total FoxO1 and p15/16 co-immunofluorescence staining in the gut of 9-month-old *tert-/-* and WT siblings. (B) Representative image of 9-month-old *tert-/-* gut. Red arrows: low nuclear FoxO1 levels in p15/16 positive cells; White arrows: high nuclear FoxO1 levels in p15/16 negative cells; scale bar: 20 μm. Dashed lines a and b depict the regions of fluorescence intensity quantification of cells analysed in D and C, respectively. (C-D) Histograms representing fluorescence quantification of DAPI, FoxO1 and p15/16 across a p15/16 positive (dashed line b) or p15/16 negative cells (dashed line a). (E) Cell analysis: High p15/16 correlates with low FoxO1 nuclear/cytoplasmic fluorescence intensity in each gut cell of *tert-/-* mutants. Analysis performed per cell basis (WT N = 3; *tert-/-* N = 2; at least 69 cells per genotype were analysed). (F) Fish analysis: On average, 9-month-old *tert-/-* fish (N = 2) contain more 'low FoxO1/high p15/16' cells than WT siblings (N = 3). Data are represented as mean per sample. p-values were calculated using a 2-factor ANOVA test.

The online version of this article includes the following source data and figure supplement(s) for figure 4:

**Source data 1.** Western Blot quantifications, as plotted in *Figure 4A*.
**Source data 2.** Mean (nuclear/cytoplasmic) p15/16 or FoxO1 fluorescence intensity per cell, as plotted in *Figure 4E*.
**Source data 3.** Mean (nuclear/cytoplasmic) p15/16 or FoxO1 fluorescence intensity per cell, as plotted in *Figure 4F*.
**Source data 4.** Western Blot quantifications, as plotted in *Figure 4—figure supplement 1–2*.
**Figure supplement 1.** Activation of AKT in older *tert-/-* mutants testis results in SOD2 reduction.
**Figure supplement 2.** Akt pathway is not induced in young tert-/-compared to wild type.

lead to increased oxidative stress and mitochondrial damage, triggering p15/16 accumulation and a senescent cell-cycle arrest.

## *tp53-/-* rescue of *tert-/-* apoptosis delays the appearance of tissue degeneration and cellular senescence

Activation of pro-proliferative signaling in an organism with defects in cell proliferation was somehow surprising. However, one major difference between 3- and 9-month-old *tert-/-* mutant gut and testis was increasing tissue damage (*Figure 1A and B*). In addition, we have previously recorded a reduction of cell divisions with aging in both *tert-/-* and WT zebrafish (*Carneiro et al., 2016b*). Thus, tissue damage may be a consequence of increased cell death in young *tert-/-* mutants that cannot be compensated by cell proliferation.

In order to maintain tissue homeostasis, dying cells in proliferative tissues induce compensatory proliferation in neighboring cells through the secretion of mitogenic signals (*Tamori and Deng, 2014*). Thus, we hypothesized that activation of the mitogenic AKT/FoxO signaling pathway was triggered to promote tissue repair in response to tissue damage in *tert-/-* zebrafish. To test this hypothesis, we rescued tissue degeneration by preventing p53 function and thereby unblock cell proliferation while restraining cell death. This way, we expected to prevent the induction of the AKT/FoxO proliferative pathway in older *tert-/-* fish and, consequently, the appearance of cell senescence.

We used *tert-/-tp53-/-* double mutant zebrafish where p53 deficiency rescues the adverse effects of telomere loss (*Anchelin et al., 2013*; *Henriques et al., 2013*). Consistent with our previous results, by preventing a p53-mediated response to telomere dysfunction, we were able to rescue the histopathological defects of older *tert-/-* testis and gut (*Figure 5A and E*). In the absence of observable tissue damage, we were no longer able to detect activation of AKT (AKT-P, *Figure 5B and F*; all genotypes N = 2), down-regulation of SOD2 (*Figure 5B and F*; all genotypes N = 2), nor accumulation of ROS (*Figure 5C and G*; all genotypes N = 3, p<0.01) in *tert-/-tp53-/-* zebrafish. Finally, consistent with our hypothesis, older *tert-/-tp53-/-* double mutant gut and testis no longer exhibit an accumulation of SA-β-Gal positive cells (*Figure 5D and H*).

Taken together, our results demonstrate that p53 is required for AKT activation and the onset of senescence in older *tert-/-* fish. Moreover, they suggest that the age-dependent switch from apoptosis to senescence is intimately linked to the loss of tissue homeostasis. In 3-month-old *tert-/-* zebrafish, telomeres are sufficiently short to trigger DDR and p53-dependent apoptosis. However, no tissue damage is observed in younger animals and this becomes apparent with age-dependent decline of cell proliferation. An older tissue with short telomeres and limited proliferative capacity responds by promoting mitogenic signaling, thereby activating the AKT/FoxO pathway, leading to mitochondrial dysfunction and senescence.

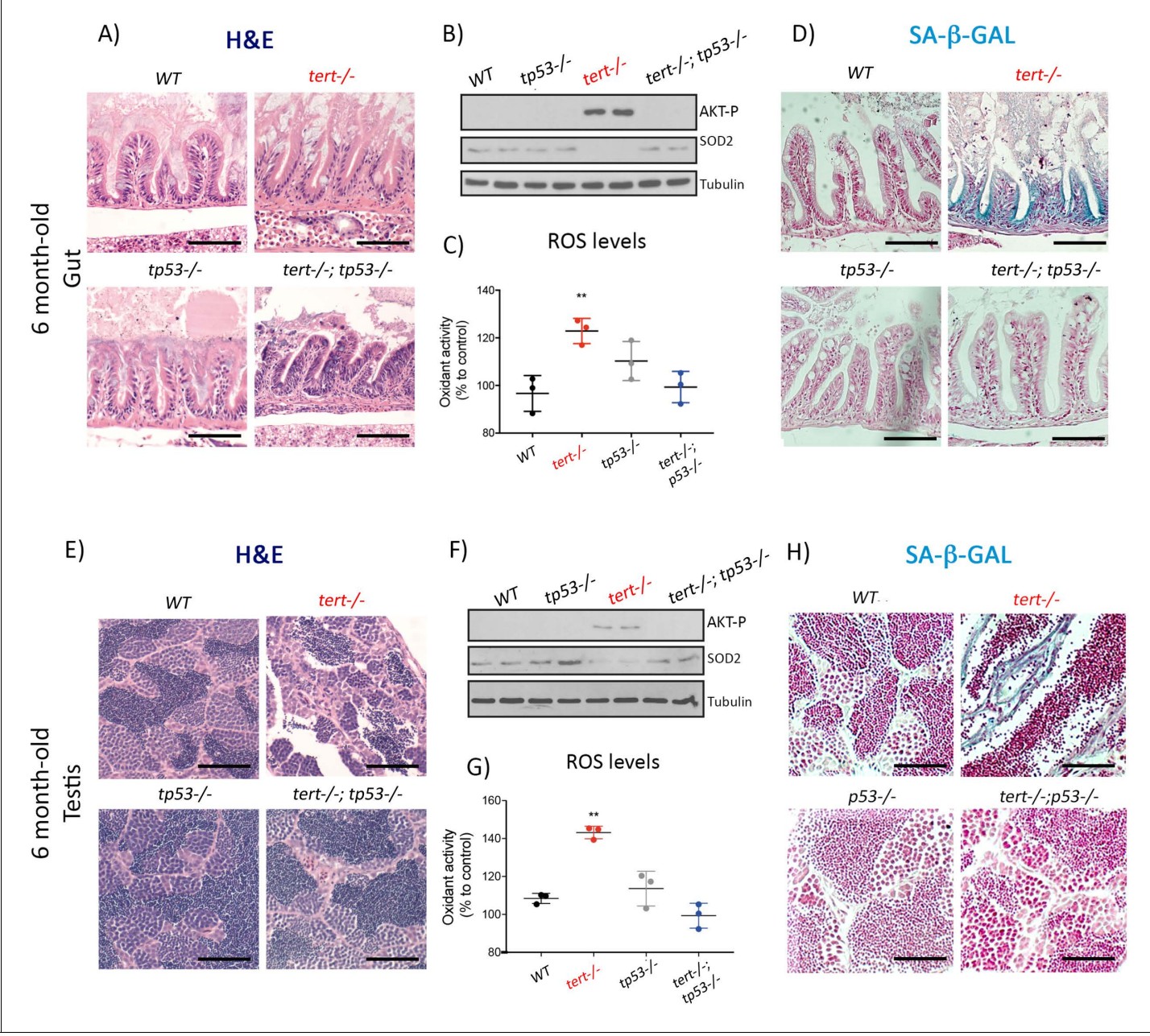

**Figure 5.** Mutation of p53 prevents short telomeres-induced tissue degeneration, Akt activation, ROS accumulation and induction of senescence. (A and E) Representative haematoxylin and eosin-stained sections of gut (A) (scale bar: 40 µm) and testis (E) (scale bar: 25 µm) from 6-month-old WT, *tert-/-*, *tp53-/-* and *tert-/- tp53-/-* siblings (N = 3 fish each);. Mutation of *tp53* in *tert-/-* fish rescues short-telomere induced tissue defects. (B and F) Representative western blot analysis of AKT-p and SOD2 in gut (B) and testis (F) (N = 2 fish each). Mutation of *tp53* in *tert-/-* fish prevents phosphorylation of AKT and downstream downregulation of SOD2 leading to a rescue of increased ROS levels (C and G; N = 3 fish per genotype). (D and H) Representative images of SA-β-GAL staining of gut (scale bar: 40 µm) (D) and testis (scale bar: 25 µm) (H) from 6 month-old WT, *tert-/-*, *p53-/-* and *tert-/- p53-/-* siblings (N = 3 fish). Data are represented as mean ± SEM (** p-value<0.01, using t-test).

The online version of this article includes the following source data for figure 5:

**Source data 1.** ROS levels measurements, as plotted in *Figure 5*.

## Inhibition of AKT activity prevents senescence in G1 and G2 *tert-/-* mutants

Our data indicate that activation of AKT in older *tert-/-* zebrafish correlates with the appearance of senescent cells. To understand the direct role of the AKT/FoxO pathway in modulating p15/16-

mediated cell-cycle arrest, we tested whether AKT activation was necessary, and therefore causal, to cell senescence. Our hypothesis would predict that inhibition of AKT phosphorylation would prevent p15/16 expression and preserve tissue homeostasis.

AKT phosphorylation is mediated by the mTORC2 complex, whose main component is the mTOR (mammalian Target Of Rapamycin) protein (*Laplante and Sabatini, 2009*). To analyze the role of AKT activation in inducing senescence upon telomere shortening, we created a double mutant bearing a mutation in the *tert* gene combined with a mutation in the mTOR zebrafish homologue (*ztor*). Previous work showed that *ztor* is essential for development and *ztor-/-* zebrafish are larval lethal (*Ding et al., 2011*). However, *ztor+/-* mutants are haploinsufficient, with the lack of one functional copy being sufficient to reduce AKT phosphorylation (*Ding et al., 2011*). Thus, we tested our hypothesis in *tert-/-ztor+/-* mutant zebrafish. As expected, 13-month-old *tert-/-ztor+/-* present reduction in AKT phosphorylation compared to *tert-/-* single mutants (*Figure 6A*, all genotypes N = 3). Consistent with our hypothesis, we observed that lower AKT activation is accompanied by a ~ 2 fold reduction of p15/16 expression (*Figure 6B*; all genotypes N = 3; p<0.01). Hence, preventing full activation of AKT in *tert-/-ztor+/-* double mutants appears to be sufficient to reduce age-associated senescence (*Figure 6A–B*; all genotypes N = 3; p<0.05). However, given the incomplete nature of zTOR inhibition, haploinsufficiency for *ztor* in a *tert-/-* mutant background was insufficient to fully restore the tissue morphology defects in *tert-/-* testis and gut (*Figure 6—figure supplement 1*). Our data corroborates previous reports describing that disruption of zTOR partially inhibits AKT activation (*Ding et al., 2011*), resulting in partial repression in p15/16 expression and, therefore, unable to completely rescue tissue morphology of older *tert-/-* mutants.

Given the previous incomplete AKT inhibition in older *tert-/-* zebrafish, we attempted a chemical inhibition in *tert-/-* fish with very short telomeres. Second-generation telomerase-deficient zebrafish (G2 *tert-/-*), obtained from incross of young homozygous *tert-/-* mutants, recapitulates several phenotypes of older G1 *tert-/-* fish (*Anchelin et al., 2013*; *Henriques et al., 2013*). G2 *tert-/-* have morphological defects, along with extremely short telomeres and lifespan. Consistent with the phenotypical recapitulation of older G1 *tert-/-* mutants, we observed that G2 *tert-/-* exhibited a marked increase of senescence revealed by SA-β-Gal staining and expression of the senescence marker p15/16 (*Figure 6C–D*; WT N = 4–5, *tert-/-* N = 4–5, pools of 10 larvae; p<0.05). Similar to older G1 *tert-/-* zebrafish, analysis of G2 *tert-/-* larvae showed that increase of senescence by p15/16 expression is concomitant with increased AKT phosphorylation, decreased SOD2 and, consequently, increase of ROS species (*Figure 6E–F*; WT N = 4, *tert-/-* N = 4; pools of 10 larvae). Our data indicate that G2 *tert-/-* larvae recapitulates aging-associated AKT activation and senescence observed in old telomerase-deficient fish.

In order to test our hypothesis that Akt-dependant reduction of mitochondrial SOD2 and consequent increase of oxidative damage are the key triggers for *tert-/-* dysfunction, we used G2 *tert-/-* larvae viability as an assay for organism defects generated by short telomeres. We prevented ROS build up by exposing G2 *tert-/-* larvae to ROS scavengers (N-Acetyl-L-Cysteine, NAC and Mito-Tempo) in growing medium starting at 3dpf, the time when ROS increase becomes apparent. In both situations and for different periods of exposure (NAC 6-10dpf and MitoTempo 3-5dpf), viability of treated G2 *tert-/-* larvae was extended beyond the untreated controls (NAC: N >= 31 fish per condition, p=0.001; MitoTempo: N >= 26; p=0.025; *Figure 6G* and *Figure 6—figure supplement 2*). Our results show that increased ROS levels in G2 *tert-/-* affect survival and that ROS scavengers are sufficient to sustain mitochondrial functions enough to prolong G2 *tert-/-* survival. Thus, even though mitochondria dysfunction is a consequence of telomere shortening, it is likely that this process is initiated, at least in part, by a reduction of mitochondrial OxPhos defenses.

Finally, we used the G2 *tert-/-* model to assess the possibility of a direct link between AKT activation and an increase in cell senescence. For this purpose, we tested whether direct AKT kinase inhibition would be sufficient to prevent p15/16 expression. For this purpose, we treated *tert-/-* and WT larvae with an AKT inhibitor (AKT 1/2 kinase inhibitor, Santa Cruz) for 2 days, once a day (*Figure 6H*). At six dpf, larvae were collected and analyzed for AKT activation and expression of senescence markers. Treatment with the inhibitor reduced AKT phosphorylation and, as a consequence, decreased p15/16 protein and mRNA levels when compared to untreated controls (*Figure 6I–J*; WT N = 3–4, *tert-/-* N = 3–4; pools of 10 larvae; p<0.05). Our data show a direct link between AKT kinase activation and senescence in *tert-/-* zebrafish. Taken together our results

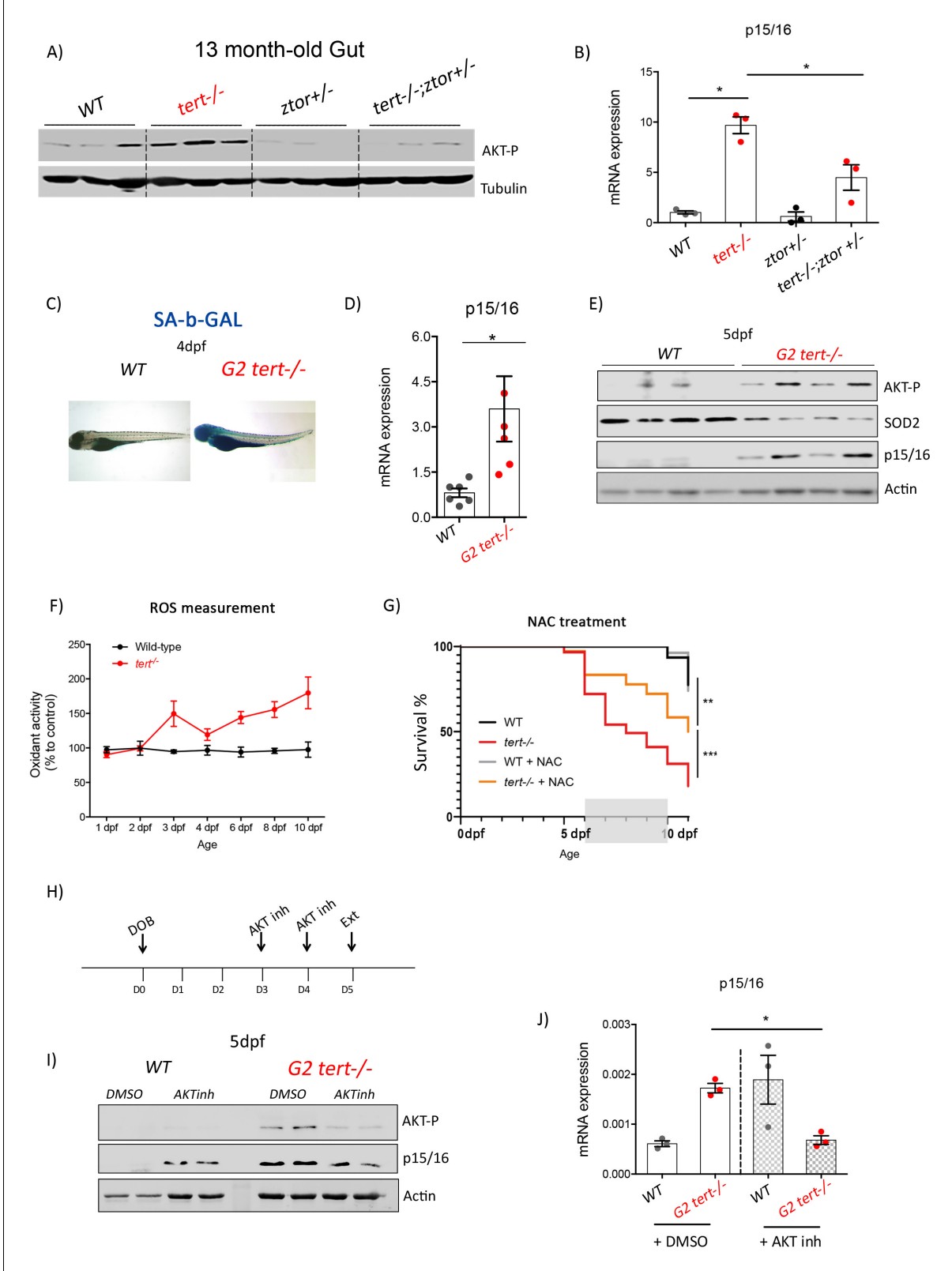

**Figure 6.** Genetic and pharmacological inhibition of AKT prevents short telomere-induced senescence. (**A**) Heterozygous mutation of zTOR counteracts telomere-shortening-induced Akt activation, leading to inhibition of p15/16 expression. Western blot analysis of AKT-P and (**B**) RT-qPCR analysis of p15/16 mRNA levels in 13-month-old gut of WT, *tert-/-, ztor+/-and tert-/- ztor+/-* fish (N = 3 fish). (**C-F**) Second generation (G2) *tert-/-* mutant larvae with extremely short telomeres show phenotypes associated with premature aging, as described in *Figures 1*, *2* and *3*. (**C**) Representative images of SA-β-
*Figure 6 continued on next page*

*Figure 6 continued*

GAL staining of WT and second generation (G2) *tert-/-* mutant four dpf larvae. (D) RT-qPCR analysis of p15/16 mRNA levels (N = 6), E) Western blot analysis of AKT-P, SOD2, p15/16 (N = 4) and (F) ROS levels measurements determined by DCFDA assay (N = 3). G) Survival curve of G2 tert-/-upon NAC (40 µM from day 6 to 10) treatment (WT N = 31; WT+NAC N = 27; G2 *tert-/-* N = 61; G2 *tert-/-* +NAC N = 36 fish; ** p-value<0.01; ** p-value<0.01 using Log-Rank test). (H-J) Pharmacological inhibition of AKT rescues telomere-shortening induced p15/16 expression. (H) Experimental scheme of pharmacological inhibition of AKT in G2 *tert-/-*. (I) Western blot analysis of AKT-P and p15/16 and (J) RT-qPCR analysis of p15/16 mRNA levels of G2 *tert-/-* and WT treated with AKT inhibitor. All RT-qPCR graphs are representing mean ± SEM mRNA fold increase after normalisation to *rpl13a* gene expression levels (* p-value<0.05; ** p-value<0.01, using t-test).

The online version of this article includes the following source data and figure supplement(s) for figure 6:

Source data 1. Real-time qPCR data of p15/16, as plotted in *Figure 6B*.
Source data 2. Real-time qPCR data of p15/16, as plotted in *Figure 6D and J*.
Source data 3. ROS levels measurements, as plotted in *Figure 6F*.
Source data 4. Survival analysis upon NAC treatment, as plotted in *Figure 6G*.
Figure supplement 1. ztor haploinsufficiency is not sufficient to suppress tissue defects in *tert-/-* zebrafish.
Figure supplement 2. Pharmacological administration of ROS scavengers increases survival of G2 *tert-/-*.
Figure supplement 2—source data 1. Survival analysis upon MitoTempo treatment, as plotted in *Figure 6—figure supplement 2*.

indicate that the telomere shortening-associated premature senescence is dependent on the activation of the AKT/FoxO pathway.

## Discussion

Homeostasis in multicellular organisms depends on coordinated responses to the external and internal insults that threaten tissue integrity. Loss of tissue homeostasis is a hallmark of aging, resulting in pathologies often caused by defective or deregulated tissue damage responses (*Neves et al., 2015*). In proliferative tissues, homeostasis relies on a controlled balance between cell proliferation and apoptosis or senescence. Telomere attrition and DNA damage are major factors contributing to aging (*López-Otín et al., 2013*). When reaching a critical length, short telomeres trigger DDRs and p53-dependent cell cycle arrest, eventually culminating in apoptosis or replicative senescence (*Francisco and Blackburn, 2001*; *Harley et al., 1990*; *Olovnikov, 1973*; *Shay and Wright, 2000*). Most cell types seem to be capable of both cellular outcomes upon damage (*Campisi and d'Adda di Fagagna, 2007*), but the molecular mechanism determining cell fate (apoptosis and senescence) in multicellular organisms remains unclear.

In the present study, we describe that young (3-month-old) telomerase deficient zebrafish already exhibit active DDRs and p53 activation. At this stage, apoptosis is the predominant cell fate. Even though DNA damage is present in proliferative tissues, such as gut and testis, no signs of cell senescence could be detected. However, there is an evident switch between apoptosis to senescence in older *tert-/-* fish. In these animals, senescence becomes the most prevalent cellular response, exhibiting increased p21, p15/16 and SA-β-Gal levels. This observation underscores the fact that the same cell types can undergo different cellular fates in vivo depending on the animal's age.

p53 transcription factor is described as a 'master regulator' of several cellular processes, including cell cycle arrest, apoptosis, senescence and autophagy (*Farnebo et al., 2010*). p53 was first shown to trigger apoptosis in response to cellular stress (*Vogelstein et al., 2000*). However, it is now acknowledged that p53 modulates genes involved in senescence depending on the stress inflicted or cell type (*Murray-Zmijewski et al., 2008*). In our study, early p53 activation in *tert-/-* zebrafish does not alter mitochondria function. However, both gut and testis of older *tert-/-* zebrafish show mitochondrial dysfunction accompanied by significant reduction of ATP levels and accumulation of ROS. These alterations are concomitant with the onset of cell senescence. However, in contrast to the previous study in mice (*Sahin et al., 2011*), we do not detect a downregulation of PGC1α neither on mRNA nor protein levels. Even though p53 is required for the older *tert-/-* zebrafish phenotypes, our results suggest that the observed mitochondrial dysfunctions are independent of PGC1α alterations.

Our study reveals that mitochondrial defects are associated with a reduction in mitochondrial OxPhos defenses. We observe that, with age, SOD2 expression is reduced in response to AKT-dependent activation and FoxO1 and FoxO4 phosphorylation. The anti-proliferative p53 and pro-survival mTOR/AKT pathways interact in a complex manner. Depending on the context, the interaction of these pathways modulates cell fate into either cell-cycle arrest, apoptosis or senescence (*Erol, 2011*; *Hasty et al., 2013*). Cell line studies show that p53 itself can inhibit mTOR/AKT pathway through several mechanisms including AMPK and PTEN activation (*Hasty et al., 2013*). p53 activation of cell senescence relies on mTOR/AKT pathway activity (*Davaadelger et al., 2016*; *Jung et al., 2019*; *Kim et al., 2017*; *Miyauchi et al., 2004*; *Vétillard et al., 2015*). Conversely, AKT inhibition reduces p53-dependent senescence (*Davaadelger et al., 2016*; *Duan and Maki, 2016*; *Kim et al., 2017*). In addition, AKT mediates the inhibition of pro-apoptotic factors (*Davaadelger et al., 2016*) and leads to increased levels of anti-apoptotic Bcl-XL (*Jones et al., 2000*; *Li et al., 2017b*). Thus, activation of mTOR/AKT pathway can act as a negative regulator of apoptosis (*Davaadelger et al., 2016*; *Duan and Maki, 2016*). AKT was also shown to induce senescence and cell-cycle arrest by elevating the intracellular levels of ROS or activating p16 transcription through direct phosphorylation of the Bmi repressor (*Imai et al., 2014*; *Li et al., 2017a*; *Liu et al., 2012*; *Miyauchi et al., 2004*; *Nogueira et al., 2008*). Consistent with these in vitro data, we showed that AKT activation in aged *tert-/-* mutant zebrafish is concomitant with increased anti-apoptotic Bcl-XL and pro-senescence p15/16 and p21 levels.

What constitutes the mechanistic nature of the switch from apoptosis to senescence? Even though young *tert-/-* mutants present no observable tissue defects, they exhibit higher levels of apoptosis and a reduction in proliferative capacity (*Carneiro et al., 2016b*; *Henriques et al., 2013*). High apoptosis increases the demand for cell proliferation from surrounding cells in a process termed apoptosis-induced compensatory proliferation (*Fan and Bergmann, 2008*). In face of cell proliferation restrictions, tissue degeneration becomes apparent in aging *tert-/-* zebrafish. In tissues where stem cells are not readily available or where tissue-intrinsic genetic programs constrain cell division, cellular hypertrophy represents an alternative strategy for tissue homeostasis (*Losick et al., 2013*; *Tamori and Deng, 2013*).

We propose that, upon telomere shortening and p53 activation, loss of tissue integrity triggers the AKT-dependent pro-proliferative pathway (*Figure 7*). The combination of these antagonistic forces in the cell would result in cellular senescence. We tested this hypothesis on both pathways. By genetically ablating *tp53* function, we were able to rescue tissue degeneration and avoid AKT activation, increased ROS levels and induction of senescence. On a second level, we inhibited mTOR/AKT genetically by dampening the *ztor* pathway and, chemically, by directly inhibiting AKT in G2 *tert-/-* larvae. In both cases, we were able to reduce the effects of telomere shortening. Collectively, our results show that the crosstalk between the two pathways, telomere shortening/p53 and AKT/FoxO signalling, regulates the apoptosis-to-senescence switch and contributes to tissue homeostasis in vivo.

Our previous studies describe a similar transition from apoptosis to senescence in WT zebrafish. While apoptosis is triggered at early stages of WT aging, loss of tissue homeostasis appears from 18 to 24 month of age and is associated with predominant senescence (*Carneiro et al., 2016b*). We therefore anticipate that the mechanisms occurring in normal zebrafish aging and leading to degenerative phenotypes are analogous to *tert-/-* premature aging. However, we were unable to observe critical telomere shortening in some WT tissues, such as testis or kidney marrow (the fish hematopoietic organ), owing to continuous expression of telomerase throughout life (*Carneiro et al., 2016b*). In this regard, *tert-/-* zebrafish are better models of human telomeropathies, a disease spectrum that affects primarily organs that rely on telomerase expression, such as the hematopoietic tissues (*Opresko and Shay, 2017*). Nevertheless, these organs do exhibit degenerative phenotypes and accumulation of senescent cells with age. Thus, other factors could trigger age-dependent dysfunctions rather than telomere shortening in WT aging. Alternatively, other organs in an aging animal, such as the immune system, may become limiting and give rise to systemic deficiencies that lead to a concerted demise.

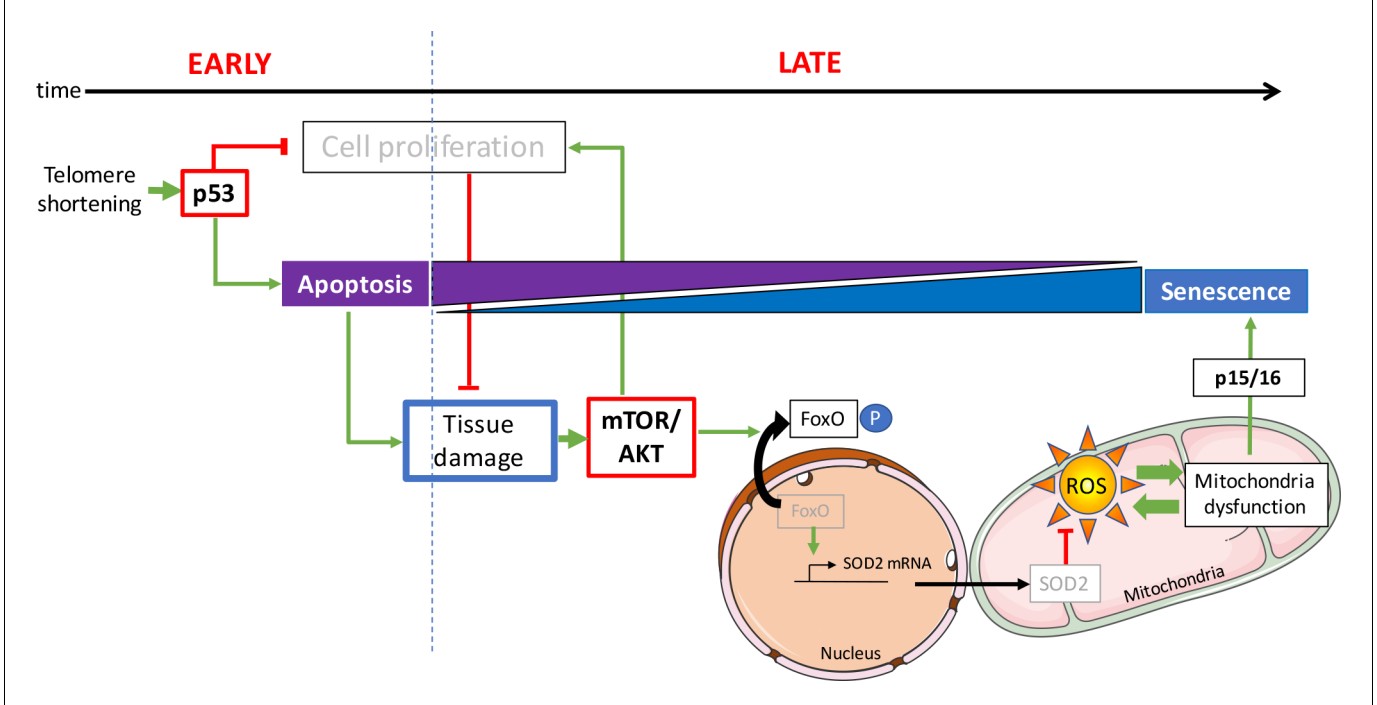

**Figure 7.** Opposing activities of p53 and mTOR/AKT upon telomere shortening promote a switch from apoptosis to senescence. Early telomere shortening triggers p53-dependent apoptosis and inhibition of cell proliferation. At early age, apoptosis is the predominant cell fate and it mostly counterbalanced by compensatory proliferation of neighboring cells. However, inhibition of cell proliferation results in a progressive loss of tissue cellularity, eventually leading to tissue damage. As age progresses, loss of tissue homeostasis triggers the pro-proliferative mTOR/AKT pathway. Akt phosphorylates FoxO, inducing its translocation from the nucleus to the cytoplasm. Loss of FoxO transcriptional activity reduces mitochondrial SOD2 expression generating mitochondria oxidative stress through increased ROS levels. Mitochondrial dysfunction eventually triggers p15/16 expression and senescence becomes the predominant cell fate.

# Materials and methods

**Key resources table**

| Reagent type (species) or resource | Designation | Source or reference | Identifiers | Additional information |
|---|---|---|---|---|
| Genetic reagent (*Danio rerio*) | *tert hu3430/+* | Hubrecht Institute, Utrecht, Netherland | RRID:ZFIN_ZDB-GENO-100412-50 | |
| Genetic reagent (*Danio rerio*) | *tp53 M214K/+* | *Berghmans et al., 2005* | RRID:ZDB-ALT-050428-2 | |
| Genetic reagent (*Danio rerio*) | *ztor xu015Gt/+* | *Ding et al., 2011* | RRID:ZDB-ALT-120412-1 | |
| Gene (*Danio rerio*) | *tert* | | ZDB-GENE-080405–1 | |
| Gene (*Danio rerio*) | *tp53* | | ZDB-GENE-990415–270 | |
| Gene (*Danio rerio*) | *ztor* | | ZDB-GENE-030131–2974 | |
| Gene (*Danio rerio*) | *cdkn2a/b* (p15/16) | | ZDB-GENE-081104–306 | |
| Gene (*Danio rerio*) | *cdkn1a* (p21) | | ZDB-GENE-070705–7 | |
| Gene (*Danio rerio*) | *bcl2l1* (Bcl-XL) | | ZDB-GENE-010730–1 | |

*Continued on next page*

*Continued*

| Reagent type (species) or resource | Designation | Source or reference | Identifiers | Additional information |
|---|---|---|---|---|
| Gene (*Danio rerio*) | *ppargc1a* (PGC1a) | | ZDB-GENE-080505–1 | |
| Gene (*Danio rerio*) | *rpl13a* | | ZDB-GENE-030131–168 | |
| Antibody | anti-p16 (mouse monoclonal; F-12) | Santa Cruz Biotechnology | #Sc-1661; RRID:AB_628067 | IF(1:50), WB (1:700) |
| Antibody | anti-FoxO1 (rabbit monoclonal; C29H4) | Cell Signaling Technology | #2880; RRID:AB_2106495 | IF(1:50) |
| Antibody | anti-zebrafish p53 (rabbit polyclonal) | Anaspec | #55342; RRID:AB_2287635 | WB (1:1000) |
| Antibody | anti-zebrafish γH2AX (rabbit polyclonal) | GeneTex | #GTX127342; RRID:AB_2833105 | WB (1:1000) |
| Antibody | anti-SOD2 (rabbit polyclonal) | Sigma-Aldrich | #SAB2701618; RRID:AB_2833106 | WB (1:1000) |
| Antibody | anti-phospho-Akt, Ser473 (rabbit monoclonal; D9E) | Cell Signaling Technology | #4060; RRID:AB_2315049 | WB (1:1000) |
| Antibody | anti-total-Akt (rabbit polyclonal) | Cell Signaling Technology | #9272, RRID:AB_329827 | WB (1:1000) |
| Antibody | anti-phospho-FoxO1, Ser256 (rabbit polyclonal) | Cell Signaling Technology | #9461; RRID:AB_329831 | WB (1:100) |
| Antibody | anti-tubulin (mouse monoclonal; B-5-1-2) | Sigma | #T6074; RRID:AB_477582 | WB (1:5000) |
| Antibody | Alexa Fluor 568 goat anti-mouse (goat polyclonal) | Invitrogen | #A11004; RRID:AB_2534072 | IF (1:500) |
| Antibody | Alexa Fluor 488 goat anti-rabbit (goat polyclonal) | Invitrogen | #A11008; RRID:AB_143165 | IF (1:500) |
| Antibody | HRP- anti-rabbit (goat polyclonal) | Santa Cruz | #Sc2054; RRID:AB_631748 | WB (1:5000) |
| Antibody | HRP- anti-mouse (goat polyclonal) | Santa Cruz | #Sc2005; RRID:AB_631736 | WB (1:5000) |
| Sequence-based reagent | *cdkn2a/b* (p15/16) Fw | This paper | PCR primers | GAGGATGAACTGACCACAGCA |
| Sequence-based reagent | *cdkn2a/b* (p15/16) Rv | This paper | PCR primers | CAAGAGCCAAAGGTGCGTTAC |
| Sequence-based reagent | *bcl2l1* (Bcl-XL) Fw | This paper | PCR primers | GGGCTTGTTTGCTTGGTTGA |
| Sequence-based reagent | *bcl2l1* (Bcl-XL) Rv | This paper | PCR primers | AGAACACAGTGCACACCCTT |
| Sequence-based reagent | *cdkn1a* (p21) Fw | This paper | PCR primers | CAGCGGGTTTACAGTTTCAGC |
| Sequence-based reagent | *cdkn1a* (p21) Rv | This paper | PCR primers | TGAACGTAGGATCCGCTTGT |
| Sequence-based reagent | *ppargc1a* (PGC1a) Fw | This paper | PCR primers | CTGTGGAACCCCAGGTCTGAC |
| Sequence-based reagent | *ppargc1a* (PGC1a) Rv | This paper | PCR primers | ACTCAGCCTGGGCCTTTTGCT |
| Sequence-based reagent | *rpl13a* Fw | *Henriques et al., 2013* | PCR primers | TCTGGAGGACTGTAAGAGGTATG |
| Sequence-based reagent | *rpl13a* Rv | *Henriques et al., 2013* | PCR primers | AGACGCACAATCTTGAGAGCAG |

*Continued on next page*

*Continued*

| Reagent type (species) or resource | Designation | Source or reference | Identifiers | Additional information |
|---|---|---|---|---|
| Sequence-based reagent | *cdkn2a/b* (p15/16) Morpholino | This paper | morpholino | TCAGTTCATCCT CGACGTTCATCAT |
| Sequence-based reagent | Control Morpholino | GeneTools | morpholino | CCTCTTACCTCAG TTACAATTTATA |
| Commercial assay or kit | In Situ Cell Death Detection Kit, Fluorescein | Roche | 11684795910 | |
| Commercial assay or kit | CellTiter-Glo Luminescent Cell Viability Assay | Promega | G7570 | |
| Chemical compound, drug | 2′,7′-Dichlorofluorescin diacetate (DCFDA) | Sigma Aldrich | D6883 | |
| Chemical compound, drug | AKT1/2 kinase inhibitor | Santa Cruz | sc-300173 | |
| Chemical compound, drug | N-Acetyl-L-Cysteine (NAC) | Sigma Aldrich | A7250 | |
| Chemical compound, drug | MitoTEMPO | Sigma Aldrich | SML0737 | |
| Other | DAPI stain | Sigma | D9542 | (0.5 µg/mL) |

## Ethics statement

All Zebrafish work was conducted according to Portuguese (Decreto-Lei 113/2013) and European (Directive 2010/63/EU) legislations and approved by the Ethical Committee of the IGC (Instituto Gulbenkian de Ciência; approval number: A001.2012) and the DGAV (Direcção Geral de Alimentação e Veterinária, Portuguese Veterinary Authority; approval number: 010294).

## Zebrafish lines and maintenance

Zebrafish were maintained in accordance with Institutional and National animal care protocols. The telomerase mutant line *tert AB/hu3430* was generated by N-Ethyl-N-nitrosourea (ENU) mutagenesis (Utrecht University, Netherlands; *Wienholds and Plasterk, 2004*). *tert AB/hu3430* line is available at the ZFIN repository, ZFIN ID: ZDB-GENO-100412–50, from the Zebrafish International Re-source Center—ZIRC. The protocols used for outcrossing mutagenized male zebrafish were previously described (*Carneiro et al., 2016a*; *Henriques et al., 2013*). All stocks were kept in heterozygous form and maintained strictly by outcrossing to AB strains to avoid haploinsufficiency effects in the progeny. The *tert hu3430/hu3430* homozygous mutant (referred in this work as *tert-/-*) was obtained by incrossing our *tert AB/hu3430* (*tert+/-*) strain. WT siblings were used as controls. Overall characterization of *tert-/-* and WT zebrafish was performed in F1 animals produced by *tert+/-* incross. Due to a male sex bias in our crosses, that affected mostly *tert-/-* progeny, we were unable to obtain significant numbers of females for analysis and so all of our data is restricted to males.

*p53* mutant line zdf1 (*tp53 M214K*, *Berghmans et al., 2005*) was kindly provided by Dr António Jacinto (CEDOC, chronic diseases, Nova medical school, Lisbon Portugal). *tp53* mutant line zdf1 (*P53 M214K*) is available at the ZFIN repository, ZFIN ID: ZDB-ALT-050428–2 from the Zebrafish International Re-source Center—ZIRC. The *tp53 M214K/M214K* homozygous mutant (referred to as *tp53-/-*) was obtained by incrossing our *tp53 AB/M214K* strain. *ztor* line was obtained from the ZFIN repository, ZFIN ID: ZDB-ALT-120412–1 from the Zebrafish International Re-source Center—ZIRC. The line was previously described (*Ding et al., 2011*) as homozygous larval lethal and it was maintained through outcrossing. All animals showing signs of morbidity that persisted for up to 5 days, such as inability to eat or swim, or macroscopic lesions/tumors were sacrificed using 200 mg/L of MS-222 (Sigma,MO, USA).

## Histological analysis

Zebrafish were sacrificed by anaesthetic overdose in 200 mg/L of MS-222 (Sigma, MO, USA), fixed for 72 hr in 10% neutral buffered formalin and decalcified in 0.5M EDTA for 48 hr at RT. Whole fish were then paraffin-embedded and three micrometer midline sagittal sections were stained with

haematoxylin and eosin for histopathological analysis. Sections were examined by a histopathologist (TC), blinded to experimental groups and microphotographs were acquired in a Leica DM2500 microscope coupled to a Leica MC170 HD microscope camera. At least four animals from each age group/genotype were analysed.

## Immunofluorescence (IF) assays

Apoptosis and Senescence was detected using the In Situ Cell Death Detection Kit (Roche, SW) according to manufacturer's instructions combined with immunofluorescence against p15/16 senescence-associated marker. Briefly, deparaffinized slides were incubated with 40 µg/mL Proteinase K in 10 mM Tris-HCl pH 7.4, 45 min at 37°C. Slides were left to cool down for 30 min at RT, washed three times in $dH_2O$ for 5 min each and blocked for 1 hr at RT in 1% BSA, 0,5% Tween 20 in PBS-T (Triton 0.5%). Subsequently, the slides were incubated overnight with anti-p16 (F-12) (1:50, Santa Cruz Biotechnology, sc-1661), followed by 3 × 10 min PBS washes. Incubation with the secondary antibody Alexa Fluor 568 goat anti-mouse (Invitrogen, UK, 1:500 dilution) overnight at 4°C was followed by 3 × 10 min PBS washes. The day after, slides were washed 2 × 5 min in PBS and then incubated with TUNEL labelling mix (protocol indicated by the supplier). Washing and mounting were performed by DAPI staining (Sigma, MO, USA) and mounting with DAKO Fluorescence Mounting Medium (Sigma, MO, USA).

For FoxO-1 and p15/16 co-staining, deparaffinized slides were incubated in Citrate Buffer (2.95 g/L sodium citrate; pH 6) for 20 min at 105°C. Samples were processed as above except for the block solution of 0.5% Triton/5% Normal Goat Serum/PBS and an overnight staining with anti-p16 (F-12) (1:50, Santa Cruz Biotechnology, sc-1661) and anti-FoxO-1 (C29H4) (1:50, Cell Signalling Technology, 2880). A second incubation was then performed using secondary antibodies Alexa Fluor 568 goat anti-mouse (Invitrogen, UK, 1:500 dilution) and Alexa Fluor 488 goat ant-rabbit (Invitrogen, UK, 1:500 dilution).

Images were acquired on a commercial Nikon High Content Screening microscope, based on Nikon Ti equipped with a Andor Zyla 4.2 sCMOS camera, using the a 20 × 1.45 NA objective, DAPI + GFP fluorescence filter sets and controlled with the Nikon Elements software.

For quantitative and comparative imaging, equivalent image acquisition parameters were used. The percentage of positive nuclei was determined by counting a total of 500–1000 cells per slide, 63x amplification (N >= 3 zebrafish per time point/genotype).

## Senescence-associated β-galactosidase assay

β-galactosidase assay was performed as previously described (*Kishi et al., 2008*). Briefly, sacrificed zebrafish adults were fixed for 72 hr in 4% paraformaldehyde in PBS at 4°C and then washed three times for 1 hr in PBS-pH 7.4 and for a further 1 hr in PBS-pH 6.0 at 4°C. β-galactosidase staining was performed for 24 hr at 37°C in 5 mM potassium ferrocyanide, 5 mM potassium ferricyanide, 2 mM MgCl2 and 1 mg/mL X-gal, in PBS adjusted to pH 6.0. After staining, fish were washed three times for 5 min in PBS pH seven and processed for de-calcification and paraffin embedding as before. Sections were stained with nuclear fast red for nuclear detection and images were acquired in a bright field scan (Leica, APERIO).

## Immunoblot analysis

Age- and sex-matched adult zebrafish fish were sacrificed in 200 mg/L of MS-222 (Sigma, MO, USA) and portions of each tissue (gonads and gut) were retrieved and immediately snap-frozen in dry ice. 4dpf larvae were sacrificed in ice and collected in 1,5 mL Eppendorf tube, minimum 10 larvae/tube. Gonads tissues and larvae were then homogenized in RIPA buffer (sodium chloride 150 mM; Triton-X-100 1%; sodium deoxycholate 0,5%; SDS 0,1%; Tris 50 mM, pH = 8.0), including complete protease and phosphatase inhibitor cocktails (Roche diagnostics), with the help of a motor pestle. Protein extracts were incubated on ice for 30 min and centrifuged at 4°C, 13.000 rpm, for 10 min. The supernatant was collected and added to 100 mL of protein sample buffer containing DTT.

Gut samples were homogenized in TRIzol (Invitrogen, UK) by mashing each individual tissue with a pestle in a 1.5 mL Eppendorf tube. After incubation at RT for 10 min in TRIzol, chloroform extractions were performed. The organic phase was collected, and proteins were precipitated according to the manufacture protocol. The protein pellet was resuspended in 100 µL of Lysis Buffer (150 mM

NaCl, 4% SDS, 50 mM Tris-HCl pH 8.0, 10 mM EDTA, complete protease and phosphatase inhibitor cocktails-Roche diagnostics).

For each sample, a fraction of proteins was separated on 12% SDS-PAGE gels and transferred to Immobilon PVDF membranes (Millipore). The membranes were blocked in 5% milk or 5% BSA (depending on the primary antibody), then incubated with the indicated primary antibody prior to incubation with the appropriate HRP-conjugated secondary antibody. Antibody complexes were visualised by enhanced chemiluminescence (ECL). Antibodies concentration: anti-p53 (1:1000, Anaspec, 55342); anti-g-H2AX (1:1000, GeneTex, GTX127342); anti-p16 (F-12) (1:700, Santa Cruz Biotechnology, sc-1661); anti-SOD-2 (1:1000, Sigma, SAB2701618); anti-phospho-AKT, Ser473 (1:1000, Cell Signaling, #4060); anti-total-AKT (1:1000 Cell Signaling, #9272, gift of Adrien Colin), anti-phospho-FoxO1, Ser256 (1:100, Cell Signaling, #9461); anti-Tubulin (1:5000, Sigma, T 6074).

## Real-time quantitative PCR

Age- and sex-matched fish were sacrificed in 200 mg/L of MS-222 (Sigma, MO, USA) and portions of each tissue (gonads, gut and muscle) were retrieved and immediately snap-frozen in liquid nitrogen. Similarly, larvae were sacrificed and collected in Eppendorf tubes, minimum 10 larvae each. RNA extraction was performed in TRIzol (Invitrogen, UK) by mashing each individual tissue with a pestle in a 1.5 mL eppendorf tube. After incubation at room temperature (RT) for 10 min in TRIzol, chloroform extractions were performed. Quality of RNA samples was assessed through BioAnalyzer (Agilent 2100, CA, USA). Retro-transcription into cDNA was performed using a RT-PCR kit NZY First-Strand cDNA Synthesis Kit # MB12501 (NZYtech).

Quantitative PCR (qPCR) was performed using iTaq Universal SYBR Green Supermix # 1725125 (Bio-Rad) and an ABI-QuantStudio 384 Sequence Detection System (Applied Biosystems, CA, USA). qPCRs were carried out in triplicate for each cDNA sample. Relative mRNA expression was normalized against *rpl13a* mRNA expression using the DCT method. Primer sequences are listed in *Supplementary file 1*.

## Detection of intracellular oxidant activity

Reactive oxygen species (ROS) accumulation was assessed by measuring the levels of the oxidized form of the cell-permeant 5-chloromethyl-2',7'-dichlorodihydrofluorescein diacetate (DCFDA, Sigma). Briefly, zebrafish were euthanized with 200 mg/L of MS-222 (Sigma, MO, USA) and tissues such as the testis, gut and muscle were dissected. Each tissue was homogenized in 100 μL of ROS buffer (0.32 mM sucrose, 20 mM hepes, 1 mM MgCl2 and 0.5 mM phenylmethanesulfonyl fluoride-PMSF). Homogenates were centrifuged and 20 μL of the supernatant was transferred to a 96-well plate and incubated in 1 μg/mL of DCFDA for 30 min. Fluorescence values were measured with a Victor three plate reader (Perkin Elmer) and normalized to total protein content, which was determined by the Bradford method. N = 3 per time point.

## ATP measurement

Age- and sex-matched adult zebrafish fish were sacrificed in 200 mg/L of MS-222 (Sigma, MO, USA) and portions of each tissue (gonads, gut and muscle) were retrieved and immediately snap-frozen in dry ice. Each tissue was homogenised in 100 mL of 6M guanidine-HCl in extraction buffer (100 mM Tris and 4 mM EDTA, pH 7.8) to inhibit ATPases. Homogenised samples were subjected to rapid freezing in liquid nitrogen followed by boiling for 3 min. Samples were then cleared by centrifugation and the supernatant was diluted (1/50) with extraction buffer and mixed with luminescent solution (CellTiter-Glo Luminescent Cell Viability Assay, Promega). The luminescence was measured on a Victor three plate reader (Perkin Elmer). The relative ATP levels were calculated by dividing the luminescence by the total protein concentration, which was determined by the Bradford method. For Bradford assays, samples were diluted (1/50) with extraction buffer.

## Electron microscopy

For electron microscopy analysis, zebrafish tissues were processed according to *Schieber et al., 2010*. Briefly, zebrafish were fixed in 2% Paraformaldehyde, 2.5% Glutaraldehyde in 0.1M PHEM buffer for 72 hr at 4°C. Dissected tissues were then washed 3 times in 0.1 M PHEM. Tissues were transferred in 1% Osmium Tetroxide in 0.1 M PHEM for 1 hr fixation on ice. Samples were then

dehydrated before being processed for embedding using Epon (*Schieber et al., 2010*). 70 nm ultra-thin sections were cut using Reichert Ultramicrotome. After being counterstained with uranyl acetate and lead, samples were analyzed using a transmission electron microscope (Hitachi H-7650).

## Larva chemical treatments

AKT1/2 kinase inhibitor (AKT inh) was purchased from Santa-Cruz (sc-300173). Stock solutions were prepared in DMSO. AKT inh was applied, after a titration, at 2 µM concentration in E3 embryo medium between 3-5dpf. Larvae were grown at 28°C and over the incubation periods, replacing the medium with an inhibitor every day. Since the inhibitor was dissolved in DMSO, controls were treated with the correspondent dilution of the solvent. The inhibitor was tested in two independent trials. Finally, 5dpf larvae were sacrificed and collected to perform protein and RNA analysis.

Second generation (G2) *tert-/-* survival experiments were performed as follows. Larvae were raised in petri dishes containing E3 embryo medium at 28°C. N-Acetyl-L-Cysteine (NAC) and Mito-TEMPO were purchased from Sigma-Aldrich. NAC was applied at 40 µM between days 5 and 10 post fertilization. MitoTEMPO was applied at 10 µM between days 3 and 5. Medium was replaced every day. Each drug experiment was performed in two independent replicates.

## Knock-down experiments using p15/16 Morpholino injection

One-cell stage WT embryos were injected with 2.4 ng or 3.6 ng of p15/16 mRNA specific translation blocking antisense morpholino oligonucleotides (MO, Gene Tools, USA) sequence (5' TCAGTTCA TCCTCGACGTTCATCAT 3') or 3.6 ng of standard control MO (5' CCTCTTACCTCAGTTACAATTTA TA 3'). After four dpf, larvae were collected for further p15/16 protein expression analysis.

## Statistical and image analysis

Image edition was performed using Adobe Photoshop CS5.1 Statistical analysis was performed in GraphPad Prism5, using one-way ANOVA test with Bonferroni post-correction for all experiments comparing WT and *tert-/-* over time. For real-time quantitative PCR, statistical analysis was performed in GraphPad Prism5, one-way ANOVA with Bonferroni post-correction. A critical value for significance of $p < 0.05$ was used throughout the study. For Western Blot the bands intensities were calculated using FIJI. Statistical analysis was performed using GraphPad Prism6, the significance was assigned according to the Mann-Whitney test. A critical value for significance of $p < 0.05$ was used throughout the study. For survival analysis, Log-rank tests were performed using GraphPad Prism5 in order to determine statistical differences of survival curves.

# Acknowledgements

We thank members from the Telomeres and Genome Stability and the Telomere Shortening and Cancer Laboratories for helpful discussions. We thank MC Carneiro and M Figueira for their help on survival experiments and A Maouche for histology guidance. We thank AR Araújo for critically reading the manuscript. We thank the Instituto Gulbenkian de Ciência histology unit, the IGC imaging unit, AL Sousa, and EM Tranfield from the IGC Electron Microscopy Facility for assistance with experimental planning, sample processing and data collection and the IGC Fish Facility for excellent animal care. IGC Fish Facility is financed by Congento LISBOA-01–0145-FEDER-022170, co-financed by FCT (Portugal) and Lisboa2020, under the PORTUGAL2020 agreement (European Regional Development Fund). MEM is a recipient of a postdoctoral fellowship from the Ville de Nice. This work was supported by the FCT (PTDC/SAU-ORG/116826/2010), Fondation Arc pour la Recherche sur le Cancer (PJA20161205137) and the Howard Hughes Medical Institute International Early Career Scientist grants received by MGF.

# Additional information

### Funding

| Funder | Grant reference number | Author |
|---|---|---|
| Fundação para a Ciência e a Tecnologia | PTDC/SAU-ORG/116826/ 2010 | Miguel Godinho Ferreira |

| | | |
|---|---|---|
| Fondation ARC pour la Recherche sur le Cancer | PJA 20161205137 | Miguel Godinho Ferreira |
| Université Côte d'Azur - Académie 4 | Installation Grant: Action 2 - 2019 | Miguel Godinho Ferreira |
| Howard Hughes Medical Institute | IECS | Miguel Godinho Ferreira |
| Ville de Nice | Postdoctoral fellowship | Mounir El Maï |

The funders had no role in study design, data collection and interpretation, or the decision to submit the work for publication.

## Author contributions

Mounir El Maï, Marta Marzullo, Inês Pimenta de Castro, Conceptualization, Data curation, Formal analysis, Validation, Investigation, Visualization, Methodology, Writing - original draft, Writing - review and editing; Miguel Godinho Ferreira, Conceptualization, Data curation, Formal analysis, Supervision, Funding acquisition, Validation, Visualization, Writing - original draft, Project administration, Writing - review and editing

## Author ORCIDs

Mounir El Maï (ID) https://orcid.org/0000-0002-7528-2474
Marta Marzullo (ID) https://orcid.org/0000-0001-7229-1693
Inês Pimenta de Castro (ID) https://orcid.org/0000-0002-0898-6834
Miguel Godinho Ferreira (ID) https://orcid.org/0000-0002-8363-7183

## Ethics

Animal experimentation: All Zebrafish work was conducted according to National Guidelines and approved by the Ethical Committee of the Instituto Gulbenkian de Ciência and the DGAV (Direcção Geral de Alimentação e Veterinária, Portuguese Veterinary Authority). Approval number: 010294.

## Decision letter and Author response

Decision letter https://doi.org/10.7554/eLife.54935.sa1
Author response https://doi.org/10.7554/eLife.54935.sa2

## Additional files

### Supplementary files

• Supplementary file 1. List of primers used in RT-qPCR expression analysis. Table listing the oligonucleotides used as primers for the RT-qPCR performed in this study.

• Transparent reporting form

### Data availability

All data generated or analysed during this study are included in the manuscript and supporting files.

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
