## [Decision Letter]

**Acceptance summary:**

We are excited how this work, by harnessing the experimental opportunity posed by a vertebrate model system of compressed aging, sheds light on the age-dependent balance between apoptosis and cellular senescence in vivo, opening several important follow-up questions and opportunities for interventions.

**Decision letter after peer review:**

Thank you for submitting your article "AKT/FoxO pathway activation switches apoptosis to senescence in short telomere zebrafish" for consideration by *eLife*. Your article has been reviewed by three peer reviewers, one of whom is a member of our Board of Reviewing Editors, and the evaluation has been overseen by Jessica Tyler as the Senior Editor. The following individual involved in review of your submission has agreed to reveal their identity: Marco Demaria (Reviewer #3).

The reviewers have discussed the reviews with one another and the Reviewing Editor has drafted this decision to help you prepare a revised submission.

Summary:

El-Mai et al. report a connection between prolonged DNA damage signaling over time and activation of cellular senescence in a zebrafish model lacking telomerase activity.

This zebrafish line presents critically short telomeres and previous studies have shown it to be short-lived and recapitulate some aspects of aging. Critically short telomeres induce DNA damage responses, which while in young zebrafish testis and intestines lead to senescence, in 9-months-old mutant zebrafish lead to a higher signal associated with cellular senescence. Subsequent compensatory proliferation associated AKT signaling leads to reduced SOD2 levels and thus increased ROS. This in turn leads to activation of p16/15 and senescence. This effect is rescued by inhibiting AKT activity pharmacologically and reducing p53 signalling leads to lower DNA damage responses and less senescence, indicated by SA-β-Gal staining. The authors demonstrate that apoptosis is p53-mediated, while the senescence phenotype depends on activation of AKT and FoxO1/4 inactivation. To note, the study provides compelling evidence for the use of zebrafish for the characterization of cell fates in vivo, particularly relative to the age of the organism. This is a straightforward analysis that provides initial in vivo mechanistic glimpses into the switch from apoptosis to senescence during aging.

The manuscript is interesting, and the concept novel. However, there are a few major considerations to take into account.

Essential revisions:

1) It's not clear whether the cell types that undergo apoptosis in three-month-old zebrafish are the same that undergo senescence in the nine-month-old zebrafish mutant lines? Can the authors deconvolve cellularity in testis and intestine between three and nine-month-old mutant zebrafish?

2) How confident are authors regarding staining/detection of senescence markers? The staining in Figure 1 seems to show an overwhelming amount of p16+ cells, particularly in testis, but panel D and F disagree with this. Quantifications are needed and additional controls and markers should be added.

3) In Figure 2, ATP levels are shown. However, ratios ADP/ATP would be more indicative of the energetic capacity.

4) From the data provided, it remains unclear if the authors are claiming that mitochondrial dysfunctions lead to excessive ROS production or vice versa. Additional experiments should be performed on this point.

5) Currently, the evidence that FoxOs mobilize from the nucleus is very poor. Additional experimental evidence should be provided in support.

6) *p53-/-* zebrafish might not be able to generate senescence at all. Authors should aim at reducing or knocking-out p53 levels in somehow older fish, as this strategy should fail to prevent senescence.

7) The paper bases a large portion of its results on semi-quantitative assays, such as Western blots, which clearly show high variability among samples (e.g. Figure 1F, p15/16, Figure 3A, P-FoxO1, P-FoxO4).

---

## [Author Response]

Essential revisions:1) It's not clear whether the cell types that undergo apoptosis in three-month-old zebrafish are the same that undergo senescence in the nine-month-old zebrafish mutant lines? Can the authors deconvolve cellularity in testis and intestine between three and nine-month-old mutant zebrafish?

We agree and we have now provided evidence that apoptosis and senescence occur in the same cell types of young and older *tert-/-* zebrafish. Since zebrafish antibodies recognizing specific cell types are lacking, we deconvolved cellularity using histology parameters, such as tissue localization, cell morphology, nucleus shape, chromatin condensation. We observed that the majority of cell types showing high levels of TUNEL in 3-month-old *tert-/-* are enterocytes in the gut and A and B spermatogonia in testis. These same populations were characterized by high levels of p15/16 immunofluorescence staining in older *tert-/-* zebrafish. These results were added as Figure 1—figure supplement 3 and commented in the manuscript.

2) How confident are authors regarding staining/detection of senescence markers? The staining in Figure 1 seems to show an overwhelming amount of p16+ cells, particularly in testis, but panel D and F disagree with this. Quantifications are needed and additional controls and markers should be added.

We acknowledge that the images provided in the original manuscript for p15/16 staining, in particular for 9-month-old testis, could raise doubts about the specificity of the staining. The poor quality of the images reflects *tert-/-* testis atrophy and tissue degeneration at the very later stages. For that reason, we excluded 9-month-old images where testis atrophy is too severe that would affect our analysis. In addition, as tissue defects are not synchronous across the populations, we included 6-month-old testis, thus considering a 6/9-month-old population. We have now provided new images for p15/16 immunofluorescence in Figure 1D.

In addition, as requested by reviewers, we performed quantification of p15/16 positive cells and TUNEL positive cells in WT and *tert-/-* zebrafish. We would like to thank the reviewers for this suggestion, as it allowed us to bring to light unexpected data. Our quantification indeed confirmed a decrease of apoptosis and an increase of senescent cells in older *tert-/-* zebrafish. Remarkably, we also observed the appearance of signs of aging in 9-month-old WT animals, as revealed by the increase of p15/16 positive cells in both gut and testis. Zebrafish live, on average, for 3.5 years and our result may reveal the very initiation of senescence associated defects in natural aging. These quantifications are now added to Figure 1E and F and commented in the manuscript.

3) In Figure 2, ATP levels are shown. However, ratios ADP/ATP would be more indicative of the energetic capacity.

We characterized mitochondrial dysfunction by measuring ATP levels alone (as described in Passos et al., 2007 and other examples in the literature) and confirmed it using electron microscopy imaging depicting mitochondria with abnormal morphologies in both gut and testis of older *tert-/-* zebrafish. We acknowledge that ratios ADP/ATP constitute a more accurate measure of the energetic capacity of cells. However, considering the long-term experiment (9 months) and the current lockdown caused by the pandemic, we were unable to perform this experiment.

4) From the data provided, it remains unclear if the authors are claiming that mitochondrial dysfunctions lead to excessive ROS production or vice versa. Additional experiments should be performed on this point.

This is a very interesting and difficult point to resolve conclusively. We agree with the reviewers that we did not provide sufficient evidence for either direction. However, while elevated ROS are described as originating mostly from mitochondrial dysfunction (Mahaseth and Kuzminov, 2017; Yang and Hekimi, 2010), these two phenomena could be self-maintained via a positive feedback loop. We have now modified our model (Figure 7) to include this cycle. However, our results indicate a down-regulation of mitochondria OxPhos defenses (SOD2) and so we would expect a consequent increase of mitochondria ROS (that, in turn, would lead to mitochondrial damage).

We acknowledge that by rescuing one of the components, one would expect to dampen the other component’s defect. Despite this caveat, we have attempted to test our hypothesis directly reducing ROS by exposing second generation (G2) *tert-/-* zebrafish to ROS scavengers (NAC and MitoTEMPO). As a result, we observed an increase in the lifespan of these very short telomere larvae. Our results suggest that mitochondria of G2 *tert-/-* larvae remain sufficiently functional to allow for increased survival when ROS levels are reduced. This result supports the hypothesis that reduction of the mitochondrial SOD2 protein levels by AKT pathway at old age of *tert-/-* increases mitochondrial ROS levels that leads to mitochondrial dysfunction. These experiments were added in Figure 6 and Figure 6—figure supplement 2 and described in the manuscript.

5) Currently, the evidence that FoxOs mobilize from the nucleus is very poor. Additional experimental evidence should be provided in support.

As requested by the reviewers, in order to provide evidence for FoxO translocation from the nucleus to the cytoplasm, we performed co-immunofluorescence staining of total FoxO1 and p15/16 in the gut of older *tert-/-* mutants. Indeed, not only we observed a reduction of FoxO1 nuclear staining, but we identified an accompanying increase of nuclear p15/16 in these cells. These differences are not present in all cells of *tert-/-* mutants, suggestive of a transition stage or microenvironment effects. Thus, in agreement with our previous data denoting increased phosphorylation of FoxO1/4 and higher p15/16 levels in old *tert-/-*, we now show they occur in the same cells. Figure 4 (previously figure 3) was adapted to include these results (Figure 4A-F) and we modified the manuscript accordingly.

6) p53-/- zebrafish might not be able to generate senescence at all. Authors should aim at reducing or knocking-out p53 levels in somehow older fish, as this strategy should fail to prevent senescence.

The reviewers are right. This experiment would be more informative than simply mutating p53 in *tert-/-* mutants from the beginning. However, knocking-out p53 in older fish requires genetic tools that are not yet available in zebrafish. Nevertheless, similar to the experiment using ROS scavengers in point 4, we analyzed the survival of G2 *tert-/-* larvae after exposure to a p53 chemical inhibitor (Pifithrin-α). Interestingly, inhibiting p53 from 3-5dpf, a time when we observe increased senescence in G2 *tert-/-* larvae, results in extension of survival similar to NAC and MitoTEMPO. These results suggest that reducing p53 activity may delay or revert the effects of senescence leading to a prolonged lifespan. However, we do not yet understand the reason for this rescue, and we decided to exclude this data from our current study. In future experiments, we will pay particular attention to a potential in vivo reversion of senescence and consequent increased proliferation of these cells.

7) The paper bases a large portion of its results on semi-quantitative assays, such as Western blots, which clearly show high variability among samples (e.g. Figure 1F, p15/16, Figure 3A, P-FoxO1, P-FoxO4).

In compliance with the reviewers’ comment, we have now provided quantification of western blots to the main figures (Figure 2 and Figure 4). These modifications allow for a better understanding of sample variability and the number of replicates used in each experiment. To solidify our observations using western blotting, we validated our data using independent assays, namely: SA-β-Gal staining (Figure 1), p15/16 immunofluorescence (and its quantification) (Figure 1), p15/16 and p21 mRNA by RT-qPCR (Figure 2). Moreover, thanks to the reviewers’ fifth major comment, we are now able to confirm the translocation of FoxO from the nucleus to the cytoplasm in old *tert-/-* fish (Figure 4). Together, our data corroborates each other supporting increased phosphorylation of FoxO in old *tert-/-* mutants.